EMBO
Molecular Medicine

# A spoonful of L-fucose—an efficient therapy for GFUS-CDG, a new glycosylation disorder

René G Feichtinger[1,†] , Andreas Hüllen[2,†], Andreas Koller[3], Dieter Kotzot[4], Valerian Grote[5], Erdmann Rapp[5,6] , Peter Hofbauer[7], Karin Brugger[1], Christian Thiel[2,†], Johannes A Mayr[1,†] & Saskia B Wortmann[1,8,*,†]

## Abstract

Congenital disorders of glycosylation are a genetically and phenotypically heterogeneous family of diseases affecting the co- and posttranslational modification of proteins. Using exome sequencing, we detected biallelic variants in GFUS (NM_003313.4) c.[632G>A]; [659C>T] (p.[Gly211Glu];[Ser220Leu]) in a patient presenting with global developmental delay, mild coarse facial features and faltering growth. GFUS encodes GDP-L-fucose synthase, the terminal enzyme in de novo synthesis of GDP-L-fucose, required for fucosylation of N- and O-glycans. We found reduced GFUS protein and decreased GDP-L-fucose levels leading to a general hypofucosylation determined in patient's glycoproteins in serum, leukocytes, thrombocytes and fibroblasts. Complementation of patient fibroblasts with wild-type GFUS cDNA restored fucosylation. Making use of the GDP-L-fucose salvage pathway, oral fucose supplementation normalized fucosylation of proteins within 4 weeks as measured in serum and leukocytes. During the follow-up of 19 months, a moderate improvement of growth was seen, as well as a clear improvement of cognitive skills as measured by the Kaufmann ABC and the Nijmegen Pediatric CDG Rating Scale. In conclusion, GFUS-CDG is a new glycosylation disorder for which oral L-fucose supplementation is promising.

**Keywords** congenital disorder of glycosylation; fucosylation; GDP-L-fucose synthase; salvage pathway; therapy
**Subject Category** Genetics, Gene Therapy & Genetic Disease

## Introduction

Some two per cent of all human genes encode for proteins involved in glycosylation, and half of all expressed proteins undergo glycosylation to achieve full functionality (Peanne et al, 2018). The field of protein glycosylation is complex and partitioned in N-glycosylation, O-glycosylation (further divided in O-fucosylation, O-glucosylation, O-GalNAcylation, O-galactosylation, O-GlcNAcylation, O-mannosylation, O-xylosylation), C-mannosylation and glycosylphosphatidylinositol (GPI)-anchored protein synthesis (van Tol et al, 2019) (Haltiwanger et al, 2015) (Takeda & Kinoshita, 1995).

The different glycosylation pathways share a joint pool of donor sugar substrates, e.g. dolichol monophosphate mannose (Dol-P-Man), uridine diphosphate glucose (UDP-Glc), uridine diphosphate galactose (UDP-Gal), uridine diphosphate N-acetylglucosamine (UDP-GlcNAc), uridine diphosphate N-acetylgalactosamine (UDP-GalNAc), guanosine diphosphate mannose (GDP-Man), guanosine diphosphate fucose (GDP-Fuc) and cytidine-5′-monophosphate-N-acetylneuraminic acid (CMP-Neu5Ac) of which most arise from the hexosamine pathway (Tuccillo et al, 2014) (de Queiroz et al, 2019). Hence, a deficiency in the allocation of one of these sugar donors can affect several glycosylation pathways. For instance, Dol-P-Man is needed in N-glycosylation, O- and C-mannosylation, just as GDP-fucose is required for the fucosylation of N- and O-glycans.

Congenital disorders of glycosylation (CDG) represent a rapidly growing group of inborn errors of protein or lipid glycosylation. In the last 10 years, the number of CDGs has more than tripled to over 130 (Ng & Freeze, 2018; Verheijen et al, 2020), with an estimated prevalence of 0.1–0.5/100,000 in Europe (Peanne et al, 2018). Clinically, CDG represent a heterogeneous disease group, with highly variable signs and symptoms, severity and age of onset.

In mammals, GDP-fucose can be synthesized (Foster & Ginsburg, 1961) de novo and via the salvage pathway (Fig 1). Usually, about

1 University Children's Hospital, Salzburger Landeskliniken (SALK) and Paracelsus Medical University (PMU), Salzburg, Austria
2 Department Pediatrics, Centre for Child and Adolescent Medicine, University of Heidelberg, Heidelberg, Germany
3 Research Program for Experimental Ophthalmology, Department of Ophthalmology and Optometry, Salzburger Landeskliniken (SALK) and Paracelsus Medical University (PMU), Salzburg, Austria
4 Clinical Genetics Unit, Salzburger Landeskliniken (SALK) and Paracelsus Medical University (PMU), Salzburg, Austria
5 Max Planck Institute for Dynamics of Complex Technical Systems, Bioprocess Engineering, Magdeburg, Germany
6 glyXera GmbH, Magdeburg, Germany
7 Department of Production, Landesapotheke Salzburg, Hospital Pharmacy, Salzburg, Austria
8 Department of Pediatrics, Amalia Children's Hospital, Radboud Center for Mitochondrial Medicine, Radboudumc, Nijmegen, The Netherlands
*Corresponding author. Tel: +43 57255 57390; E-mail: s.wortmann@salk.at
†These authors contributed equally to this work

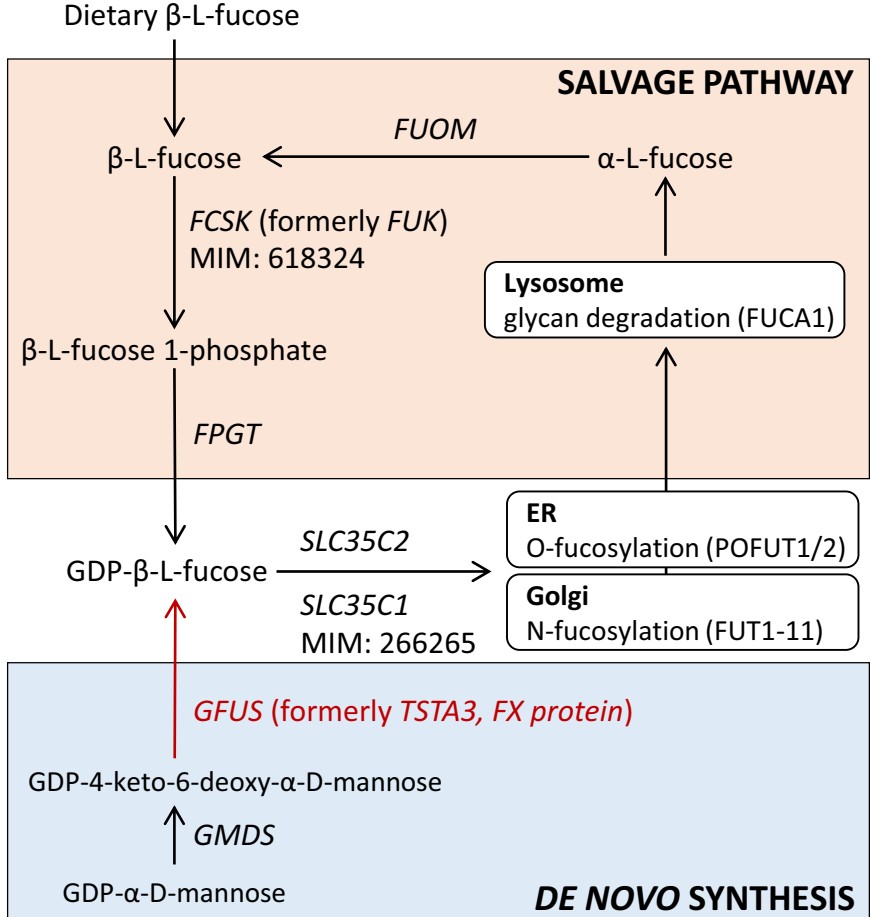

**Figure 1.  Synthesis of GDP-L-fucose by the de novo and salvage pathway.**

Shown are the two metabolic pathways for GDP-L-fucose synthesis by the salvage and *de novo* pathways. In the salvage pathway, fucose obtained from diet or lysosomal degradation is converted to GDP-fucose in two steps by L-fucose kinase (*FCSK*) followed by fucose-1-phosphate guanylyltransferase (*FPGT*). In the de novo synthesis, which is responsible for most of the GDP-fucose yielded, GDP-fucose is obtained from GDP-D-mannose. The enzymes involved are GDP-D-mannose-4,6-dehydratase (*GMDS*) and GDP-L-fucose synthase (*GFUS*). GDP-fucose is then transported to the endoplasmic reticulum (ER) and to the Golgi by transporters SLC35C2 and SLC35C1, respectively. Genes with previously described disease-causing mutations are indicated by their respective MIM numbers.

90% of GDP-fucose originates from the *de novo* synthesis (Yurchenco & Atkinson, 1977) using the nucleotide-activated sugar GDP-α-D-mannose (GDP-Man) as starting substrate. Its three-step conversion is carried out by two enzymes, the GDP-D-mannose-4,6-dehydratase (GMDS) followed by the two-step (epimerase and reductase) NADPH-dependent conversion of GDP-4-dehydro-6-deoxy-D-mannose to GDP-fucose catalysed by the GDP-L-fucose synthase (GFUS, formerly termed FX protein or TSTA3) (Tonetti *et al*, 1996). Within the salvage pathway, L-fucose originating from food or from the lysosomal breakdown of glycoproteins is directly used as a substrate (Michalski & Klein, 1999). Here, L-fucose kinase (FCSK) catalyses the first step of the salvage pathway and synthesizes fucose-1-phosphate, the donor substrate for fucose-1-phosphate guanylyltransferase (FPGT) which forms GDP-fucose (Ishihara *et al*, 1968) (Fig 1).

Next, GDP-fucose is transported from the cytosol into the ER by (presumably) SLC35C2 for O-fucosylation, which is catalysed by the fucosyltransferases POFUT1 (Wang *et al*, 2001) and alternatively by POFUT2 (Chen *et al*, 2012). Concerning fucosylation of N-glycans,

GDP-L-fucose has to be transported from the cytosol into the Golgi by the Golgi GDP-fucose transporter SLC35C1. Then, core and antenna fucosylation of N-glycans is conducted by fucosyltransferases FUT1-11 (Fig 1) (Schneider *et al*, 2017). L-fucose is mainly integrated in glycoproteins (Moloney & Haltiwanger, 1999) but can also be found in glycolipids (McKibbin, 1978). In mammalian N- and O-glycans, the deoxyhexose L-fucose is linked by α-1,2-glycosidic bonds with galactose or by α-1,3-, α-1,4-, or α-1,6-glycosidic bonds with N-acetylglucosamine. In case of O-fucosylation, fucose is bound to serine or threonine residues of nascent proteins.

Fucose plays an important role in intra- and intercellular recognition processes. The ABO and Lewis X blood group systems are determined, among others, by this sugar. Depending on the type and location of the binding of the fucose to the basic structure of the blood group antigens, substances with H, Lewis A or Lewis B specificity arise. Furthermore, glycans with α-1,3-linked fucose molecule are important for the selectin-mediated rolling of leukocytes on the surface of the capillary endothelial cells in the context of

lymphocyte recirculation and inflammatory processes (Becker & Lowe, 2003). Fucosylated glycoproteins are also essential for fertilization (Johnston *et al*, 1998) and the embryonic (Fenderson *et al*, 1990) as well as postnatal (Bendahmane & Abou-Haïla, 1997) development especially in neuronal tissue.

While for most CDG no therapeutic approach exist, increasingly therapeutic strategies by sugars or trace elements (for MPI-CDG [mannose], SLC35C1-CDG [fucose], PGM1-CDG [galactose], SLC35A2-CDG [galactose], TMEM165-CDG [galactose], SLC39A8-CDG [manganese]) are proven successful at least in individual patients (Witters *et al*, 2017; Brasil *et al*, 2018; Peanne *et al*, 2018)). Those therapies usually rely on a surplus of the respective donor substrate or cofactor for the defective enzyme.

Here we present a girl with developmental delay/intellectual disability (DD/ID), growth failure, mild coarse facial features and biallelic variants in *GFUS* leading to a defective GDP-fucose synthesis *in vivo* and *in vitro*. We show that oral fucose supplementation can restore intracellular fucosylation and significantly improve the neurological phenotype as well as the growth.

# Results

## Case report

This female was born as second female to healthy Austrian parents (Fig 2A–C). The family history was unremarkable with exception of the older sister having a concentration deficit and some problems with visual-spatial tasks, but otherwise developed age-adequately. The pregnancy was complicated by detection of small cerebral cysts which was stable during pregnancy. The neonatal ultrasound was unremarkable with no detection of cysts, as was the ultrasound at the age of 9 months. Foetal child growth was within normal limits, and foetal movements were unremarkable. A caesarean section was performed at the 38[th] week of gestation due to breech presentation. Anthropometric data at birth were within normal limits (birth weight 3,210 g (p50), birth length 52 cm (p50), birth head circumference 36 cm (p97)) and neonatal adaptation was unremarkable. From birth on lack of appetite and aversion against feeding was noticed without swallowing problems,

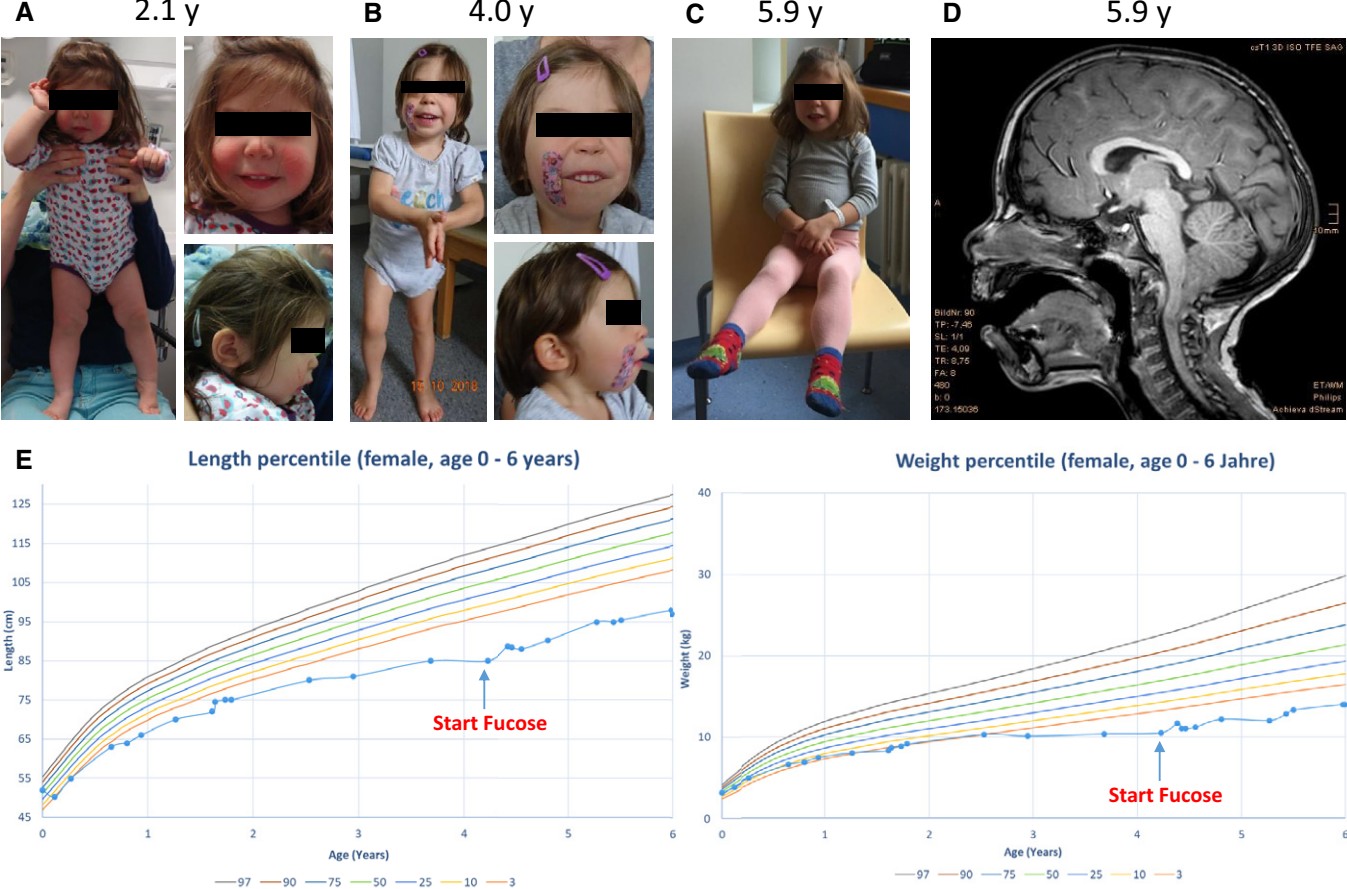

**Figure 2.  Clinical and neuroimaging findings in one individual with biallelic GFUS variants.**

A   Photograph of the individual with 2.1 years.
B   Photograph of the individual with 4 years.
C   Photograph of the individual with 5.9 years.
D   Magnetic resonance imaging (MRI) of the affected individual at the age of 5.9 years.
E   Growth charts / anthropometry of the patient before and after treatment with fucose (Kromeyer-Hauschild *et al*, 2001).

diarrhoea or vomiting. Initially, she was fed via a nasogastric tube, later via gastrostomy. Despite all efforts including different formula feedings and clinical rehabilitation programmes, no sufficient growth and weight gain could be reached. Her development was globally delayed, she learned walking by the age of 3 9/12 years. She was a cheerfully playing girl, but did not interact a lot with her environment, she was singing some syllables when playing. By the age of 4 4/12 years, it was not possible to test her with an age-appropriate test kit due to lack of interest and concentration. Alternative use of the Bayleys Scale II found a developmental age of about 23 months. The Nijmegen Pediatric CDG Rating Scale showed a total score of 10 (section I: 3, section II 0, section II 7 points (Table EV3).

An MRI at the age of 2 11/12 years revealed normal inner and outer CSF spaces and age-adequate myelination. A small corpus callosum was noted as well as an Arnold Chiari I malformation (Fig 2D). There was no clinical suspicion of seizures, and an EEG at the age of 12 months was unremarkable.

At the physical examination at the age of 4 4/12 years, proportionate short stature (height 89 cm, $P < 3$), body weight 11 kg ($P < 3$) and head circumference (48 cm, p10) were noted (Fig 2E). She had mild coarse facial features, almond-shaped eyes, slightly down slanting palpebral fissures, long philtrum, mild retrognathia as well as low implanted ears (Fig 2A–C). She had full hair with a low nuchal hairline and mild hypertrichosis on both arms. Complete blood counts and clotting studies were unremarkable. Her blood group was O Rh-positive, and no Bombay phenotype was detected. Isoelectric focussing of transferrin was unremarkable. Based on history and clinical chemical investigations (full blood counts, IgG/A/M/E and vaccination titre to regular vaccinations), no other organ involvement or immunological problem was detected.

**Biallelic variants in *GFUS***

Exome sequencing did not show variants in known disease genes, but revealed two variants in the 13 exons spanning *GFUS* gene

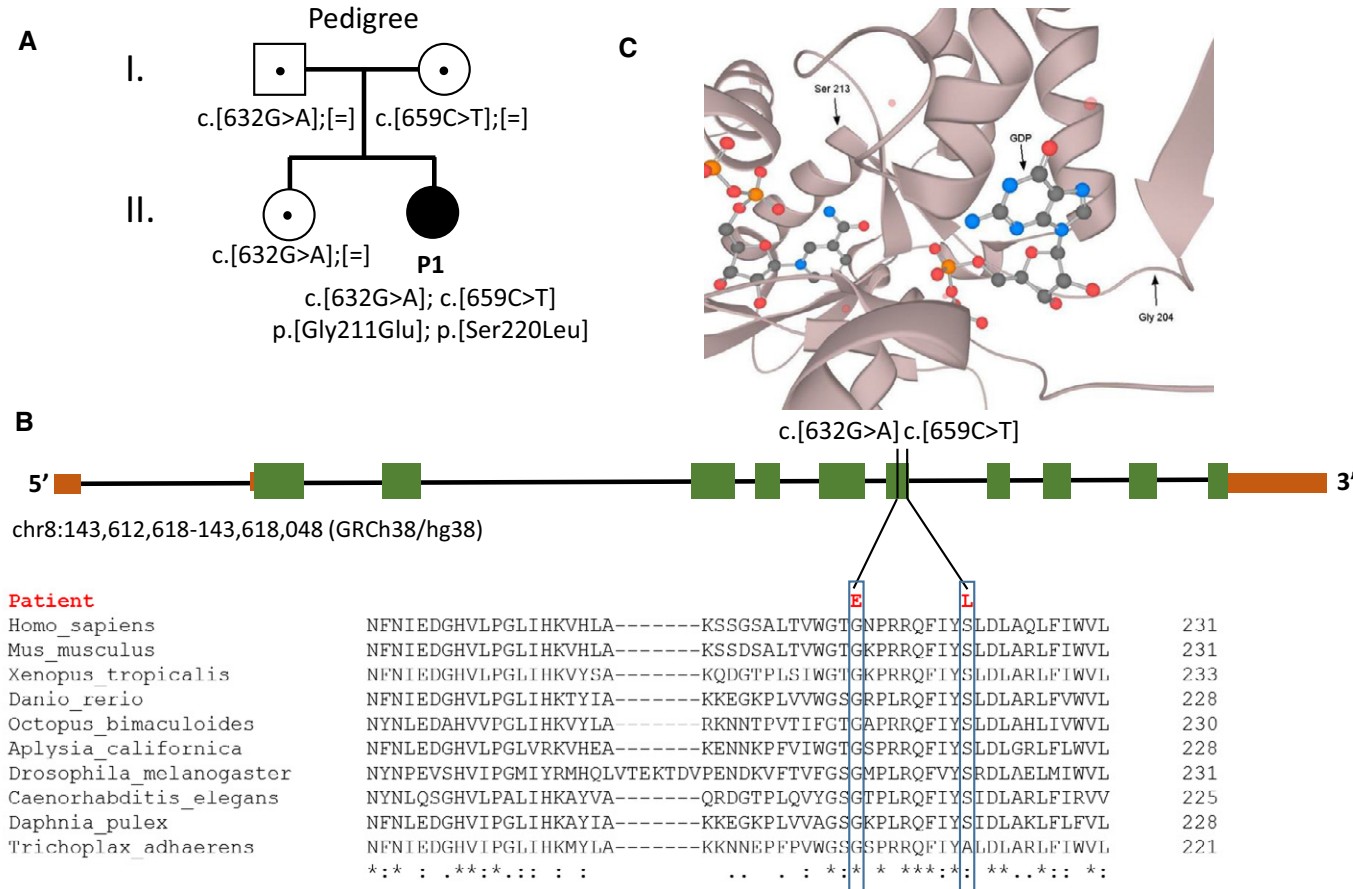

**Figure 3. Mutations identified in GFUS in one family, *GFUS* gene organization and partial structure of GFUS protein.**

A   Compound heterozygous mutations were identified in *GFUS* in one family.
B   (Top) Gene structure of *GFUS* and localization of variants. Introns length was reduced 1.5-fold. 5′ and 3′ UTR is shown in brown and the coding exons in green. (Bottom) Alignment and conservation of GFUS protein sequences from different species by Clustal Omega.
C   Partial protein structure of GFUS. Amino acid Gly204 is localized close to the GDP-D-mannose substrate-binding domain, whereas amino acid Ser213 is in neighbourhood to the binding domain of the cofactor NADPH (Zhou et al, 2013).

(NM_003313.4) located on 8q24.3 (GRCh38.p13 [GCF_000001405.39]), for which segregation analysis by Sanger sequencing showed that c.632G>A (p.Gly211Glu) was inherited paternally and c.659C>T (p.Ser220Leu) maternally (Fig 3A). The healthy older sister was found to be heterozygous for the c.632G>A variant. Both variants are rare; c.632G>A (p.Gly211Glu) is found in 2 out of 250738 alleles (allele frequency 8.0E-6) and c.659C>T (p.Ser220Leu) 3 out of 250354 alleles (allele frequency 1.2E-5) in the gnomAD database, none of them homozygous. The affected amino acid residues are highly conserved from human (NP_003304) to *Trichoplax adhaerens* and are predicted to be damaging by several prediction programmes (SIFT, PolyPhen, REVEL, MetaLR) (Fig 3B and C).

### Significantly diminished GFUS protein and mRNA expression in patient-derived fibroblasts

Western blot analysis of patient fibroblasts revealed a significantly diminished GFUS amount of $26.2 \pm 34.1\%$ (*$P = 0.0233$) when compared to controls (Fig 4A). Immunofluorescence analysis for GFUS showed a co-staining with the cytosolic PMM2. qRT–PCR studies displayed reduced quantities for the patient's *GFUS* mRNA

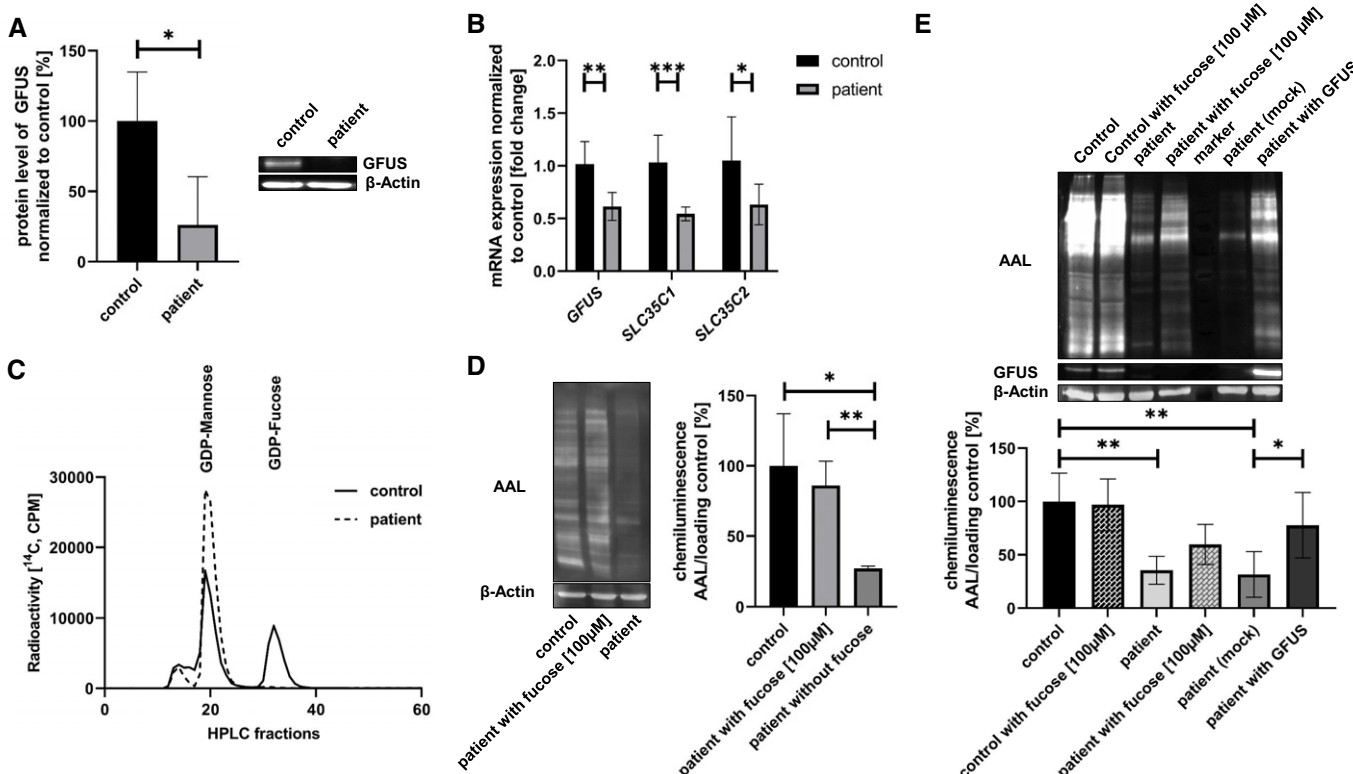

**Figure 4. Biochemical characterization of GFUS deficiency in patient-derived fibroblasts.**

A  Western blot analysis and quantification of GFUS. Expression of GFUS was analysed by western blotting on a 15% SDS–PAGE with cytosolic fractions derived from control and patient fibroblasts. Quantification revealed a significant decrease of GFUS protein. Data were obtained from fibroblasts; *n* = 4; experiment was independently repeated four times, for statistics an unpaired *t*-test was performed.

B  mRNA expression studies. qRT–PCR studies showed a significant decrease in mRNA level of *GFUS* by 40.0% (± 9.3%, **$P = 0.0010$) normalized to a control. Also, expression of *SLC35C1* and *SLC35C2* were significantly decreased in patient-derived fibroblasts. Data were obtained from fibroblasts; *n* = 10; experiment was independently repeated three times, for statistics an one-way ANOVA was performed.

C  Conversion of GDP-D-mannose into GDP-L-fucose. The conversion GDP-D-[$^{14}$C]mannose to GDP-L-[$^{14}$C]fucose was analysed in an *in vitro* assay. After 1 h, nearly half of GDP-D-mannose was transformed to GDP-fucose in the control cell line, whereas the patient's fibroblasts were nearly unable to catalyse the reaction. Data were obtained from fibroblasts; *n* = 1.

D  Lectin binding studies and quantification in fibroblasts. *Aleuria aurantia* lectin (AAL) was used to address the fucosylation level in fibroblasts. *AAL* staining was performed before and after treatment of the patient cell line with 100 μM L-fucose. Before sugar supplementation, a significant loss of *AAL* signal strength was measured, whereas after treatment a significant re-fucosylation was detected. Data were obtained from fibroblasts; *n* = 4; experiment was independently repeated four times, for statistics an unpaired *t*-test was performed.

E  Complementation study with patient-derived fibroblasts. Viral infection was used to introduce an empty cloning vector and the wild-type *GFUS* cDNA in patient cells, respectively. For analysis lysates of a control cell line, a control cell line supplemented with 100 μM fucose, patient cells, patient cells supplemented with 100 μM fucose, patient cells transfected with an empty vector and patient cells transfected with wild-type *GFUS* were analysed by AAL staining clearly indicating the disease-causing influence of the defective *GFUS* in the patient. Western blot analysis against GFUS further showed expression of wild-type GFUS protein in the infected patient-derived fibroblasts. Data were obtained from fibroblasts; *n* = 4; experiment was independently repeated two times, for statistics an one-way ANOVA was performed.

Data information: *$P < 0.05$; **$P < 0.01$; ***$P < 0.001$. Bars and error bars represent mean ± SD. Exact *P*-values are reported in the results part.

level (61.5 ± 13.3%; **$P = 0.0021$). Besides, also the mRNA expression levels for the Golgi GDP-fucose transporter *SLC35C1* (54.4 ± 6.5%; ***$P = 0.00003$) as well as for the potential ER GDP-fucose transporter *SLC35C2* (63.3 ± 17.4%; *$P = 0.035$ (Fig 4B)) were significantly diminished.

### Reduced GDP-L-fucose synthesis from GDP-D-mannose in patient-derived fibroblasts

The conversion of GDP-D-[$^{14}$C]mannose to GDP-L-[$^{14}$C]fucose was followed over 60 min. In patient-derived fibroblasts, the allocation of GDP-L-fucose was reduced to 1.15% compared to the controls which converted 45% from GDP-D-[$^{14}$C]mannose to GDP-L-[$^{14}$C]fucose within the same time frame (Fig 4C).

### Fucose supplementation leads to re-fucosylation in patient-derived fibroblasts

Patient and control fibroblasts were stained with AAL lectin. Before supplementation of the patient's cell culture medium with 100 μM L-fucose for 48 h, the fucosylation of proteins was significantly diminished to 27.05% (± 1.58%; *$P = 0.021$) in comparison to controls. L-fucose addition of 100 μM significantly increased the fucosylation level (86.08 ± 17.29%; **$P = 0.0042$) (Fig 4D).

### Retroviral complementation of patient-derived fibroblasts verifies GFUS deficiency underlying the biochemical phenotype

Both, an empty vector (mock) and the wild-type *GFUS* cDNA, were retrovirally introduced into the patient cells, respectively. Non-infected and mock-infected patient fibroblasts showed a significantly reduced level of fucosylation of 35.5% (**$P = 0.0042$) and 31.7% (**$P = 0.0025$), respectively, when compared to controls. Infection of patient cells with the wild-type *GFUS* cDNA led to expression of GFUS whereby a significantly enhanced fucosylation was induced in the patient's fibroblasts (77.8 ± 30.7%, *$P = 0.045$) (Fig 4E).

### Normal behaviour of serum transferrin during isoelectric focussing but abnormal whole serum N-glycosylation

Isoelectric focussing of the patient's serum transferrin revealed no abnormalities; therefore, we measured serum N-glycosylation by multiplexed capillary gel electrophoresis with laser-induced fluorescence detection (xCGE-LIF). In comparison to the controls ($n = 10$), the patient's N-glycome showed no shift in its relative composition. In contrast, the patient's serum N-glycans displayed a substantial reduction of fucosylated N-glycans (−18.0%). The relative distribution of core-fucosylated glycans was decreased to 10.3%, whereas in the controls a distribution of 27.9 ± 6.5% was found. In line with this, antenna-fucosylated glycans were also found to be decreased to 0.07% in the patient in comparison to the control pool (0.6 ± 0.3%) (Fig EV1A, Table EV1).

Furthermore, four core-fucosylated N-glycans (FA2BG2S2(2,6), FA2BG1[6], FA1G1S1(2,6)[3] and/or FA1G1S1(2,3)[6], FA2BG2S1 (2,6) and/or FA2G2S1(2,3)[6]) were found in control derived serum, which were absent in the patient sample. In contrast to this, five non-fucosylated N-glycans were only present in patient-derived serum (A2G0, A2G2, A2BG2, A2BG1[6], A4G0). Notably, the fucosylated structure FA2G2S1(2,6)S1(2,3) was found to be unique for the patient (Fig EV1B; Table EV1).

### Oral fucose supplementation improved lectin staining of patient serum and blood smears

Compared to controls, the patient serum showed a significantly reduced fucosylation of 36.6% (± 10.5%; **$P = 0.0026$) when analysed with AAL (Fig 5A). While the peripheral blood mononuclear cells (PBMCs) and platelets of controls and the heterozygous mother of the patient stained positive for AAL, the patients PBMCs and platelets did not even show fucosylation at baseline (Figs EV2A–L and EV4A–G). After 2 and 4 weeks of oral fucose supplementation, a positive fucosylation staining in patients PBMCs and platelets was detected (Fig EV2B2–D2 and B4–D4). Western blots of serum and PBMCs stained with AAL lectin revealed reduced levels of fucosylation in the patient compared to controls (Fig EV4B and D). A gradual increase of fucosylation was present in the patient treated with oral L-fucose from 2 to 8 weeks (Fig EV4E–G).

### Oral fucose supplementation increased fucosylation of PBMCS shown by flow cytometry analysis (FACS)

At baseline, almost no fucosylation on the surface of patient PBMCs (1.27 mean fluorescence intensity [MFI]) was detected with FACS compared to the heterozygous mother used as control (97.23 MFI) (Figs 5B and EV2E1). L-fucose (200 mM) completely blocked the cell staining of PBMCs and was comparable with the unstained sample. Therefore, the blocking of cell staining by 200 mM L-fucose served as a negative control. After 2 weeks of oral L-fucose supplementation, an increase of fucosylation of PBMCs was present (20.67 MFI) (Fig EV2E2). After 4 weeks of fucose treatment, the patient PBMCs showed almost a similar cell staining/fucosylation (167.42 MFI) as the control PBMCs (217.83 MFI), although the PBMCs of both aggregated at the day of analysis which influenced the cell staining (Fig EV2E3). Eighth weeks after the start of fucose treatment, patient PBMCs revealed an even stronger cell surface fucosylation (67.89 MFI) than control PBMCs (47.18 MFI) (Figs 5C and EV2E4).

### GFUS deficiency affects expression of downstream proteins associated with fucose metabolism

Since whole serum N-glycan analysis revealed an impairment of the core fucosylation, protein expression of FUT8 was examined by western blot which revealed a significantly elevated protein level of FUT8 (+105,7 ± 21.58%; $P = 0.0084$) in patient-derived fibroblasts compared to controls. To also address expression of O-fucosyltransferases in patient cells, POFUT1 and POFUT2 protein levels were analysed. While POFUT1 expression remained widely unchanged (−2.49 ± 6.18%), expression of POFUT2 showed a reduced signal (38.05 ± 14.05%) (Fig EV3A). Interestingly, we also found the level of ADAMTS13, a heavily O-fucosylated protein, in the patient's serum before start of the fucose therapy to be reduced (−14.1%), whereas an enhanced expression of this protein was achieved under 8 weeks of fucose (+17.3%; Fig EV3B).

Analysis of the mRNA expression revealed a significantly increased transcript level of *FUT8* (2.79 fold change; ±1.10 fold

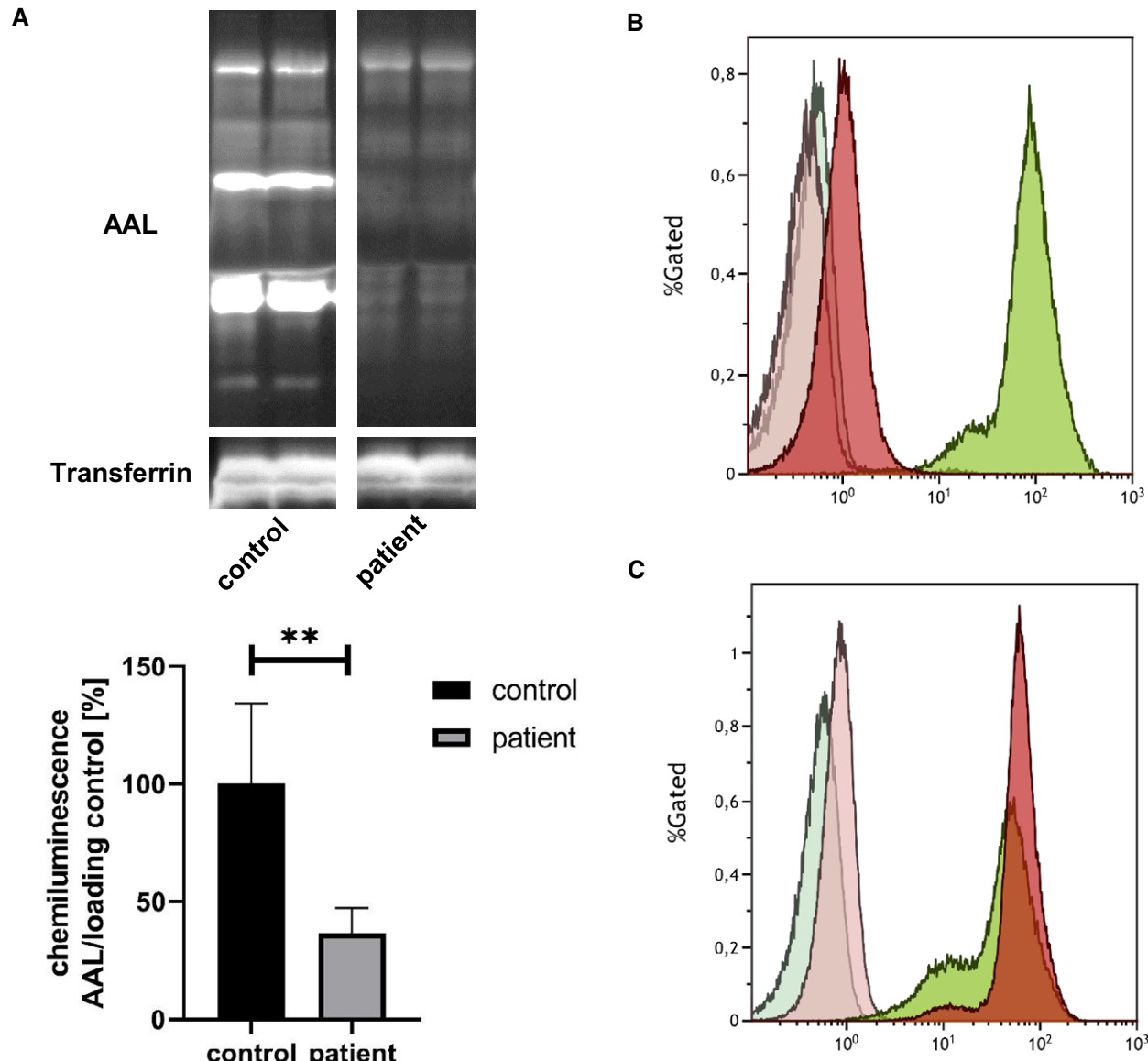

**Figure 5. *Aleuria aurantia* lectin blot of the affected individual and controls and FACS analysis.**

A   *Aleuria aurantia* lectin (AAL) was used to address the fucosylation level in sera of controls and the patient (left) which revealed significant loss of fucose residues in case of the patient. Data were obtained from serum; *n* = 6; experiment was independently repeated three times, for statistics an unpaired *t*-test was performed.

B   FACS analysis of PBMCs. Results of flow cytometry analysis of patient (red) and control (green) PBMCs stained with 1 µg/ml FAA lectin without (dark) or with 200 mM L-fucose (light) before the start of fucose therapy.

C   FACS analysis of PBMCs. Results of flow cytometry analysis of patient (red) and control (green) PBMCs stained with 1 µg/ml FAA lectin without (dark) or with 200 mM L-fucose (light) 8 weeks after therapy start

Data information: \*\*$P < 0.01$. Bars and error bars represent mean $\pm$ SD. Exact *P*-values are reported in the results part.

change; $P = 0.0498$) normalized to a control. Expression of *POFUT1* (94.4%) and *POFUT2* (102.6%) was not changed significantly.

**Clinical course after 19 months of oral L-fucose supplementation**

No side effects of oral fucose were noted. At the time of writing, the patient is aged 5 11/12 years. The follow-up MRI was unchanged. All growth parameters have stabilized ((height 98 cm, $P < 3$), body weight 14.9 kg ($P < 3$) and head circumference (49 cm, $P < 3$);

Fig 2). Without knowing about the treatment independently, the kindergarten as well as the physiotherapist report a developmental spurt. She learned to talk in two to three word sentences and shows much more interaction with her environment, e.g. while playing, and by clearly expresses and enforces her will. She has improved attention span and, e.g., now regularly choses to sit down alone at a table for drawing or handcrafting. The child learned to perform transitions much faster and has improved exercise tolerance especially while running around. She learned to eat soup with a spoon and

started to ask for food herself and accepted more variation in food; the PEG was no longer necessary and removed. She learned to (un)-dress herself, completed toilet training and learned to perform daily routines like clearing her toys in the evening independently. Psychological testing was possible with an age-adequate test kit, the Kaufmann ABC (Table EV2) where she showed a below average total IQ of 72 but with age-adequate test results in some subtests like short term memory and recognition of faces. The Nijmegen Pediatric CDG Rating Scale was improved by 3 points (total score: 7, section I: 1, section II 0, section II 6 points (Table EV3).

## Discussion

We here report the first patient with a defect in the *de novo* synthesis of the nucleotide-activated sugar GDP-L-fucose which is the donor sugar substrate for the N- and O-fucosylation of proteins. Deficiencies in the process of protein fucosylation belong to the very rare types of CDG. Among the 130 different defects known to date (Peanne *et al*, 2018; Verheijen *et al*, 2020), only FCSK-CDG, SLC35C1-CDG, FUT8-CDG and POFUT1-CDG have been reported to be disease-causing aside from GFUS-CDG in fucose metabolism (Etzioni *et al*, 1992; Marquardt *et al*, 1999; Lubke *et al*, 2001; Li

*et al*, 2013; Ng *et al*, 2018a; Ng *et al*, 2018c) (Fig 6). As GFUS, also FCSK, which affects the phosphorylation of β-L-fucose, is an enzyme needed for the allocation of GDP-fucose in the cytosol but in contrast to GFUS belongs to the salvage pathway of GDP-fucose synthesis (see Fig 1). SLC35C1-CDG (formerly CDG-IIc/ LADII) impacts the Golgi GDP-fucose transporter which transports GDP-L-fucose from the cytosol into the Golgi. FUT8 and POFUT1 are fucosyltransferases for assigning the sugar residue from GDP-L-fucose to N- and O-glycans in the Golgi and ER, respectively (Luhn *et al*, 2001; Ihara *et al*, 2007; Holdener & Haltiwanger, 2019).

Almost all of the described patients with a fucose deficiency suffer from short stature, developmental delay (DD), intellectual disability (ID), facial dysmorphisms as broad forehead and nasal bridge and recurrent infections. Additionally, seizures or epilepsy (SLC35C1, FCSK-CDG, FUT8-CDG) and immunological phenotypes (SLC35C1-CDG) have been reported. People who carry no functional H antigens on their red blood cells (e.g. due to hypofucosylation), present serologically with the Bombay (0h) blood group, whereby at least some of the SLC35C1-CDG patients can be distinguished from other patients described in the context of defects in the fucose metabolism. (Frydman *et al*, 1992; Lübke *et al*, 2001; Etzioni *et al*, 2002; Ng *et al*, 2018b; Ng *et al*, 2020; Park *et al*, 2020). An exception is POFUT1-CDG, which mainly manifests by reticulate

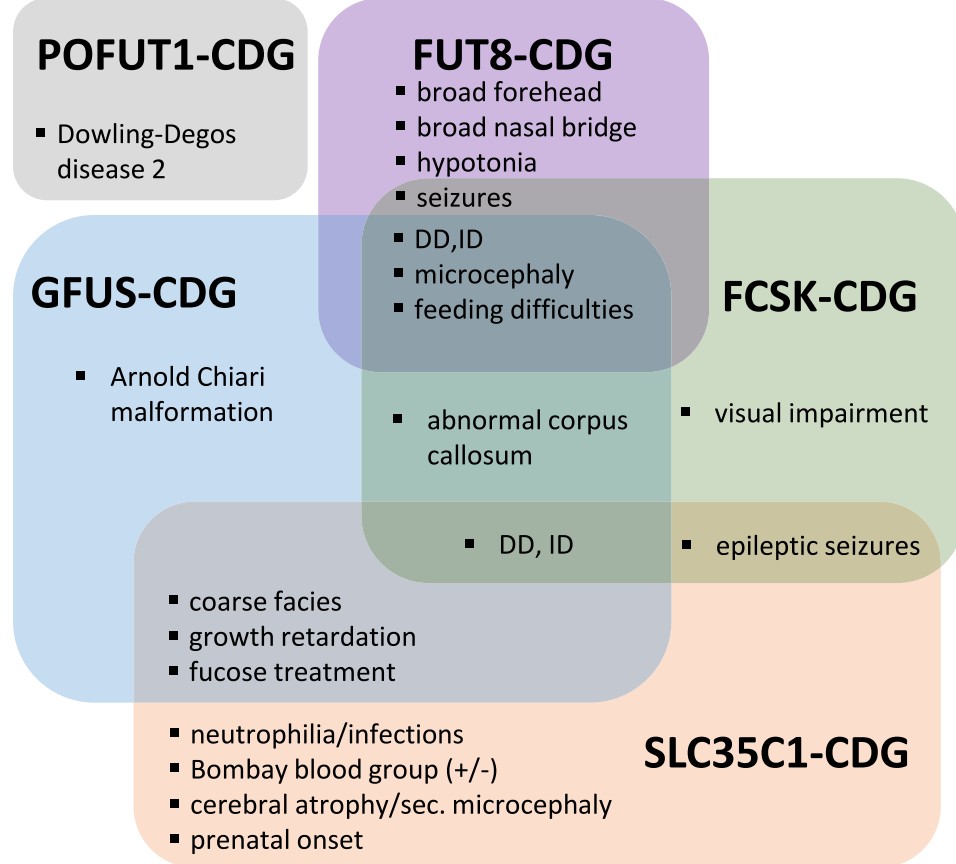

**Figure 6.  Clinical phenotypes of human diseases related to fucose metabolism.**

Symptoms of patients affected by disorders of fucose metabolism.

hyperpigmentation after puberty known as Dowling-Degos disease 2 (Li et al, 2013).

Due to the restricted number of patients, our synopsis must be rated as preliminary. Therefore, it remains to be determined whether the listed clinical data are already sufficient for the identification and classification of further patients in the future (Fig 6).

However, the neurodevelopmental phenotype of our patient is in line with the importance of fucosylated glycoproteins in the embryonic (Fenderson et al, 1990) and postnatal (Bendahmane & Abou-Haïla, 1997) development, especially in neuronal tissue. An increasing number of fucosylated glycans could be detected in healthy human brains. The fucose residues on the glycans are important for neuronal development, learning and memory (Mountford et al, 2015; Tosh et al, 2019). In a Gfus$^{-/-}$ (formerly Fx$^{-/-}$) mouse strain, most of the knockouts were not viable. Surviving mice exhibit postnatal failure to thrive which could be rescued by oral L-fucose supplementation (Smith et al, 2002), a feature that is consistent with the human phenotype. Interestingly, adult Gfus$^{-/-}$ mice developed neutrophilia and a LAD (leukocyte adhesion deficiency)-like defect comparable to what is seen in humans with SLC35C1-CDG.

The important role of fucose in development was also shown in a knockout mouse model for SLC35C1-CDG. Slc35c1$^{-/-}$ presented with, e.g. severe growth retardation, dilatation of lung alveoles and hypocellular lymph nodes. In vitro and in vivo leukocyte adhesion and rolling assays revealed a severe impairment of P-, E- and L-selectin ligand function (Hellbusch et al, 2007). Studies on brain further revealed a reduced size of the dentate gyrus in the hippocampus and elevated amounts of synapses accompanied by behavioural abnormalities as impairment of learning and memory (unpublished data of the Thiel laboratory).

Concerning the patient presented here, we speculate that her complex clinical phenotype is due to the combination of reduced fucosylation of N- and O-glycans provoked by the biallelic mutations in GFUS which led to a significant reduction on the transcript and protein level identified by qRT–PCR and western blot. Besides, visualization of crystal structure 4E5Y (Zhou et al, 2013) using Protein Data Base 3d-viewer tool indicates that variant p.Gly211Glu lies in close distance to the GDP-D-mannose substrate-binding domain, whereas variant p.Ser220Leu is assumed to be near the binding domain of the cofactor NADPH which might both result in reduced enzymatic activity. Although it was not further addressed whether only one or both variants accounted for the effects on transcript and protein level, it clearly proved a significant reduced synthesis of GDP-L-fucose as outcome in case of the patient.

Concerning the patient's N-glycans, a general lack of fucosylation in patient-derived fibroblasts, PBMCs and serum by lectin staining was detected. Since the viral complementation of patient-derived fibroblasts with the wild-type GFUS cDNA restored the observed biochemical phenotype, the disease-causing nature of the detected variants in the patient's GFUS could be approved.

In-depth N-glycan studies of whole serum proteins by xCGE-LIF showed a general hypofucosylation of sugar moieties in our patient. Notably, several unique glycans in the patient's serum were detected like the di-antennary sugar moieties A2G0, A2G2, A2BG2, A2BG1[6] and the tetra-antennary A4G0 which all could only be found as fucosylated opposites in the controls. Furthermore, the core-fucosylated N-glycan FA2G2S1(2,6)S1(2,3) could only be detected in the patient. Whether one of these mentioned N-glycans

can serve as diagnostic marker for GFUS-CDG needs to be determined in subsequent studies with additional patients.

Although both the antennae and core fucosylation were affected, we found a more prominent effect on the glycan basis in which a fucose residue is bound in alpha-1,6 orientation to GlcNAc catalysed by FUT8. Paradoxically, FUT8 transcript and FUT8 protein expression were found to be significantly increased in qRT–PCR and western blot studies. Since, among the fucosyltransferases, FUT8 emerges with its unique function in transferring the sugar residue in alpha-1,6 linkage, we rate the increased FUT8 expression as compensatory effect.

Additionally, the Golgi GDP-fucose transporter SLC35C1 showed to be significantly decreased in qPCR studies hinting to a parallel compensatory downregulation of this transporter due to the lack of its nucleotide-activated donor sugar substrate in the cytosol. As the determination of ABO and Lewis$^X$ blood group systems depend on fucosylation, some severely affected SLC35C1-CDG patients present with the Bombay blood group phenotype (Cooper et al, 2020) due to the loss of their Sialyl Lewis$^X$ structures which are normally found on granulocytes and monocytes. Notably, this was not seen in our GFUS patient. Furthermore, glycans with α-1,3-linked fucose molecule are important for the P-selectin-mediated rolling of leukocytes on the surface of the capillary endothelial cells in the context of lymphocyte recirculation and inflammatory processes (Becker & Lowe, 2003). In the first reported SLC35C1-CDG patients, neutrophilia in combination with reduced leukocyte adhesion and recurrent infections were dominating clinical findings. Although a clear effect on the core and antennae fucosylation in the patient was detected, no immunological phenotype in our GFUS patient could be seen. This could be due to the functional overlap of the salvage and the de novo pathway. In addition, enzymatic activity of the GFUS fibroblasts was not totally absent, whereby at least a minimum of GDP-L-fucose was synthesized by the de novo pathway. Many clinical symptoms of the GFUS patient are consistent with the SLC35C1-CDG and FUT8-CDG patients described so far.

Concerning the O-fucosylation pathway, a reduced transcript level of SLC35C2 was found which is assumed to encode for the ER GDP-L-Fucose transporter (Lu et al, 2010). Furthermore, although expression of POFUT1 and POFUT2 was normal in qRT–PCR studies, a reduced protein level of POFUT2 (~62%) could be detected. Even more important, an effect on ADAMTS13 (a disintegrin and metalloprotease with thrombospondin type 1 motif 13) was observed in serum when analysed by western blotting. ADAMTS13 is carrying 8 thrombospondin type 1 (TSP1) motifs which are O-fucosylated by POFUT2 in the Golgi. Since secretion of ADAMTS13 was shown to depend on the O-fucosylation of its TSP1-repeats (Ricketts et al, 2007; Lancellotti et al, 2013), an impact of diminished GFUS on this glycosylation pathway can be assumed as well. In animal embryos, the Notch signalling pathway is involved in the development of organs and tissues and, through the cell-cell interaction via lateral inhibition, significantly determines the shape of the organs. Notch function depends on N-glycosylation, O-glycosylation, O-GlcNAcylation and O-fucosylation (Takeuchi & Haltiwanger, 2014). A failure in this signal path can lead to impairment of the liver, skeleton, heart, eye, face, kidney and vasculature (Penton et al, 2012). Importantly, Notch is also a master player of neurogenesis and neuronal survival (Lasky & Wu, 2005; Imayoshi & Kageyama, 2011; Engler et al, 2018; Gómez-Pinedo et al, 2019; Santopolo et al,

2020). Notably, in a mouse model with a deletion of the *Gfus* locus, fucosylation deficiency led to suppressed Notch activation (Wang *et al*, 2017). Taken together, the broad impairment of glycosylation and signalling pathways might help to understand the complex clinical phenotype of our patient.

In humans, a salvage pathway of GDP-L-fucose synthesis exists beside the—in GFUS deficiency impaired—*de novo* synthesis pathway. Under physiological conditions about 90% of GDP-L-fucose is derived from *de novo* synthesis (Yurchenco & Atkinson, 1977). It was therefore hypothesized that oral fucose administration could improve the patient's phenotype. This was further underpinned by the observation that supplementation of the growth media with fucose improved the protein fucosylation in our patient's fibroblasts.

As fucose is able to pass the blood-brain-barrier (Harsh *et al*, 1986) and has also already been given to patients with SLC35C1-CDG and led here to some positive effects (Marquardt *et al*, 1999), a treatment with oral fucose was initiated in the GFUS patient.

Oral fucose treatment was well tolerated in our patient and no side effects were observed. There was a moderate improvement of growth parameters so far but more important a significant improvement in neurological development objictied by psychological testing. Surprisingly the patient scored within the age-adequate range in several subtests after 19 months of fucose treatment while being unable to being characterized by a regular test kit before. This is consistent with the reported role of L-fucose for synaptic plasticity, neurite outgrowth and neuron morphology and axon guidance, all important in learning, memory and cognitive processes (Pohle *et al*, 1987; Krug *et al*, 1994; Kalovidouris *et al*, 2005; Schneider *et al*, 2017) (Kondoh *et al*, 2014; Kondoh *et al*, 2017) (Koscielak *et al*, 1987).

This study is limited by the presence of only one affected patient. Therefore, it can only be speculated why it was not possible to find a second individual through gene matching initiatives and international collaboration. An explanation could be the importance of fucosylation for many human processes and could be mirrored in the low viability of Gfus$^{-/-}$ mice. In the gnomAD database, no homozygous LoFs are present for *GFUS*. Another possibility displays lacking biomarkers for the detection of fucosylation deficiencies. As fucose does not carry an electrical charge, isoelectric focussing, the regular screening test for CDGs, is normal, and due to the low shift in mass, the same accounts for analysis conducted by HPLC (Lefeber *et al*, 2011). Thus, GFUS-CDG adds to the list of glycosylation defects with normal serum transferrin IEF. Hence, patients with defects in the fucosylation pathways can only be detected by more advanced approaches, like xCGE-LIF or mass spectrometry (Pralow *et al*, 2020) in combination with genetic screening.

Given the simple and efficient treatability, GFUS-CDG turned out to be another example for the importance of early screening of DD/ID and should be kept in mind as candidate for future newborn (genetic) screening.

## Materials and Methods

The parents gave written informed consent for their child for inclusion into this study and authorized to show clinical images. All studies performed followed the declaration of Helsinki (64$^{th}$ WMA General Assembly, Fortaleza, Brazil, October 2013) and were approved by the ethics committee Salzburg (415-E/2552/10-2019). The experiments conformed to the principles set out in the Department of Health and Human Services Belmont Report.

### Genetic investigations

Proband only exome sequencing (ES) from leukocyte derived DNA was performed as described previously (Kremer *et al*, 2016a; Kremer *et al*, 2016b; Wagner *et al*, 2019). In brief, coding regions were enriched using a SureSelect Human All Exon V5 kit (Agilent) followed by sequencing as 100-bp paired-end runs on an Illumina HiSeq 4000. Reads were aligned to the human reference genome (UCSC Genome Browser build hg19) using Burrows-Wheeler Aligner (v.0.7.5 a) (Li & Durbin, 2009). Single-nucleotide variants and small insertions and deletions (indels) were detected with SAMtools (version 0.1.19). Variants based on an autosomal recessive filter were prioritized for variants with a minor allele frequency (MAF) < 0.1% and for *de novo* variants with MAF < 0.01%. Sanger sequencing was performed using standard methods using the forward 5′-CC ACGACAACTTCAACATCG-3′ and reverse 5′-CTGGAGACAGGAGC CTATGG-3′ primers.

### Sample preparation of blood smears and isolation of PBMCs

One drop (approximately 20 µl) of blood was used to prepare blood smears on Thermo Scientific™ SuperFrost Plus™ (Thermo Fisher Scientific, 15438060). Blood smears were air dried for at least 24 h prior to staining.

Serum was extracted for lectin blotting by centrifugation of EDTA blood for 10 min at 600 × *g*.

For isolation of peripheral blood mononuclear cells (PBMCs), EDTA blood was diluted 1:1 with phosphate-buffered saline (PBS). 10 ml Ficoll®-Paque Premium (Merck, 17-5442-02) was carefully overlayered with 20 ml of the 1:1 blood/PBS mixture in a 50 ml tube. The sample was centrifuged at 400 × *g* for 30 min at 20°C. The layer containing mononuclear cells was transferred into a 15-ml tube. Cells were diluted with 10 ml PBS and centrifuged at 400 × *g* for 10 min at 20°C. The supernatant was discarded, and the pellet was suspended in 1 ml PBS.

### Lectin studies in PBMCs and serum

Serum was diluted 1:10 with Tris pH 7.4 and PBMC diluted 1:3 in Tris pH 7.4. Samples were mixed with 5× sample buffer (0.02% bromophenol blue, 5% β-mercaptoethanol, 20% glycerol, 10% SDS, 250 mM Tris pH 6.8), sonicated for 10 s, heated to 95°C for 5 min and centrifuged at 13,000 *g* for 1 min. Different amounts (5, 10, 15 µl) of the lysates were loaded for better comparability of the individual and her unaffected mother. Proteins were separated on a 10% SDS–PAGE gel for 35 min at 150 V with running buffer (12.1 g Tris, 17.9 g tricine, 10 ml SDS (10%), dissolved in 1 l with ddH$_2$O). Blotting was done with the Trans-Blot Turbo System (Bio-Rad) and the following specifications: 1.3 A, 25 V, 7 min. Proteins were transferred to nitrocellulose membranes (Trans-Blot® Turbo™ Mini Nitrocellulose Transfer Packs, Bio-Rad, 1704158). Membranes were dried and incubated for approximately 1 min with 0.1% Ponceau red in 5% acetic acid. Destaining was done with ddH$_2$O until the

desired staining intensity was reached, and protein loading was determined. Membranes were washed 3× for 5 min in PBS and blocking of the membranes was performed with a 1× carbohydrate free blocking solution (Vectorlabs, Carbo-Free™ Blocking Solution, SP-5040) diluted in ddH$_2$O, 0.05% Tween-20 for 30 min. Blots were incubated for 30 min with the biotinylated *Aleuria aurantia* lectin (Vectorlabs, B-1395) at a final concentration of 10 μg/ml in 1x PBS, 0.05% Tween-20, pH 7.6. Blots were washed 3× short and 3× for 5 min with PBS-T. Samples were incubated for 1 h with Streptavidin-HRP (Cell Signaling Technology, 3999S) 1:2,000 in 1x PBS, 0.05% Tween-20, pH 7.6. Membranes were washed 3× short and 3× 5 min in PBS-T. Development was performed with the Lumi-LightPLUS Western Blotting Substrate (Roche, 12 015 200 001). Membranes were washed and incubated overnight at 4°C with rabbit anti-GAPDH (Glyceraldehyde-3-Phosphate Dehydrogenase Rabbit Polyclonal Antibody, Trevigen, 2275-PC-1) 1:5,000 in Tris-buffered saline buffer, 0.5% Tween-20, 1× western blot blocking reagent (Roche, 11 921 673 001), pH 7.5 (TBS-T). Samples were washed 3× short and 3× 5 min with 1x TBS. Detection of GAPDH was performed with an alkaline phosphatase-based system according to manufacturer instructions with the goat anti-rabbit (GAR) Intro Kit (Bio-Rad, 170-5014).

### Lectin staining of blood smears

Air-dried blood smears were fixed for 10 min in ice-cold methanol. Blood smears were washed 3 × 3 min in PBS and incubated for 30 min with fluorescein-labelled *Aleuria aurantia* lectin (AAL) (Vectorlabs, VECFL-1391). Samples were washed 3 × 3 min with PBS and incubated for 10 min with DAPI diluted 1:2,000 in PBS. Samples were washed 2 × 3 min in ddH$_2$O and mounted with fluorescence mounting medium (DAKO, S302380-2). At the next day, images were taken with a confocal laser scanning microscope.

### Cell culture

Patient and control fibroblasts were cultured in Dulbecco's modified Eagle's medium (high glucose; Life Technologies, Karlsruhe, Germany) containing 10% foetal calf serum (PAN Biotech, Aidenbach, Germany) and 1% Pen/Strep at 5% CO$_2$ at 37°C. Growth medium was changed every 72 h. For the sugar supplementation studies, L-fucose (100 μM in PBS) was added to the growth medium for 48 h. For viral infection of fibroblasts, the ecotropic packaging cell line FNX-Eco (ATCC) and the amphotropic packaging cell line RetroPack PT67 (Clontech) were cultured in DMEM containing 1 × L-glutamine, 1× Pen/Strep and 10% foetal calf serum (PAN Biotech GmbH; heat-inactivated at 65°C for 30 min) at 37°C under 5% CO$_2$.

### Production of retroviruses and infection of patient-derived fibroblasts

Ecotropic FNX-Eco cells (5 × 10$^5$) were seeded onto dishes (60 mm diameter) 24 h before transfection. Transient transfection by FuGENE6 reagent was performed according to the manufacturer's protocol (Roche), with 1 mg of pLNCX2 vector (mock) and pLNCX2 with the wild-type *GFUS* cDNA. Further procedures were performed as described elsewhere (Thiel *et al*, 2002). The supernatant with the

amphotropic retroviral particles was used to infect patient and control fibroblasts. After infection, the medium was replaced by DMEM containing 10% heat-inactivated FCS with 225 mg/ml G418 (Gibco BRL). Selection was carried out for 10 days.

### *In vitro* conversion of [$^{14}$C]GDP-D-mannose into [$^{14}$C]GDP-L-fucose

Fibroblasts of a control and the patient were plated and cultured for 48 h prior to the experiment. Cells were harvested by scrapping in PBS and homogenized by passing 15 times through a 22-gauge needle before centrifugation for 2 h at 100,000 × *g* and 4°C. The supernatant was taken as cytosol. The *in vitro* conversion experiment was performed at 37°C with 70 μg protein as previously described (Korner *et al*, 1999). [$^{14}$C(U)]GDP-D-mannose (0.2 μCi/ sample were used; specific activity = 55 mCi/mmol; concentration = 0.1 mCi/ml) was purchased from Biotrend Chemikalien GmbH (Köln, Germany).

### Western blot analysis of GFUS, FUT8, POFUT1, POFUT2 and ADAMTS13

Western blots were performed by standard procedures with 20 μg protein derived from patient's and control fibroblasts. Samples were mixed with 6× Laemmli buffer (375 mM Tris–HCl, pH 6.8, 6% SDS, 48% glycerol, 9% 2-mercaptoethanol, 0.03% bromophenol blue) and denatured for 5 min at 95°C. Extracts were analysed on a 15% SDS gel and blotted onto a nitrocellulose membrane (GE Healthcare, Munich, Germany) by semi-dry electrophoretic transfer. The membrane was blocked for 1 h at room temperature with 5% milk powder either in TBS-0.1%T (20 mM Tris, 137 mM NaCl, 0.1% Tween-20, pH 7.5) for detection using the GFUS antibody (Thermo Fisher, PA5-27473, rabbit anti-human) or PBS-T (0.1% Tween in phosphate-buffered saline [PBS]) for β-actin (antibody Sigma A5441) detection. After blocking, the membrane was washed and incubated overnight with the primary antibody in a dilution of 1:1,000 at 4°C. Washing steps were repeated three times. Membranes were incubated for 1 h at RT with secondary antibodies, either anti-rabbit IgG-conjugated with horseradish peroxidase (HRP) (for GFUS detection) or anti-mouse IgG-HRP (for β-actin; Santa Cruz, Dallas, Texas; 1:10,000) was used. Protein signals were detected by light emission with Pierce™ enhanced chemiluminescence reagent (ECL) western blot analysis substrate.

For FUT8 (Proteintech, 66118-1-lg, mouse anti-human), POFUT1 (Proteintech, 14929-1-AP, rabbit anti-human), POFUT2 (Proteintech, 17764-1-AP, rabbit anti-human) and ADAMTS13 (Thermo Fisher, MA5-34796, rabbit anti-human), the procedure was performed analogously. All these primary antibodies were used in a dilution of 1:500 and the corresponding secondary antibodies in a dilution of 1:10,000.

### qRT–PCR for *GFUS, SLC35C1, SLC35C2, FUT8, POFUT1* and *POFUT2*

For total RNA isolation, the RNAeasy kit (Qiagen) was used with 1 × 10$^6$ control and patient's fibroblasts, respectively. cDNA was reversed transcribed from 1 μg total RNA with random hexamer primers and RevertAid Reverse Transcriptase (Thermo Scientific) according to the manufacturer's guideline. Quantitative real-time

analysis was carried out with 50 ng cDNA as described (Himmelreich *et al*, 2019) with the following primer sets:

*GFUS* (NM_003313.4): Fwd: 5′-TGGACGCAGGTGCAACTGAC-3′ and Rev: 5′-CTGTGCTGTATCCGTGAGATC-3′; *SLC35C1* (NM_018389.4): Fwd: 5′-CTTCAACGTGCTGCTCTCCT-3′ and Rev: 5′-GGAGCACCTTCGTGGTGTAG-3′; *SLC35C2* (NM_173179.4): Fwd: 5′-GTTCCACCTGCAGCCACTCA-3′ and Rev: 5′-CCAGGAGGAACTCAGAGAAGC-3′; *FUT8* (NM_004480.4): Fwd: 5′-GGACACTGGTCAGGTGAAGT-3′ and Rev: 5′-AAACTGAGACACCCACCACA-3′; *POFUT1* (NM_015352.1): Fwd: 5′-CTGCCTGACCTGAAGGAGAT-3′ and Rev: 5′-ACCACCTTCACCTTCCCTTT-3′; *POFUT2* (NM_015227.4): Fwd: 5′-CGGATTCCCTGGTCTGAGTT-3′ and Rev: 5′-TTCCCAGGTCCCTTCTTTCC-3′. Primers for amplification of the reference gene *RAB7A* were *RAB7A*-Fwd: 5′-46GATTCTGGAGTCGGGAAGACAT67-3′ and *RAB7A*-Rev: 5′-235TGTAGAAGGCCACACCGAGA216-3′. Expression levels were normalized to *RAB7A* and set in relation to the results from control fibroblasts.

## Lectin blot analysis of fibroblasts

20 µg of protein per lane derived from control and patient's sera or fibroblasts were separated on a 10% SDS–PAGE and transferred onto a nitrocellulose membrane as described above. After blotting, the nitrocellulose membrane was washed three times in TBS-0.5%T (20 mM Tris, 137 mM NaCl, 0.5% Tween-20, pH 7.5) at RT, followed by an 1 h incubation in TBS-0.5%T. To the membrane, the biotinylated *Aleuria aurantia* lectin (AAL; 2 mg/ml; Vector Laboratories Inc., Burlingame, California, USA) was added (1:1,000 in TBS-T) and incubated overnight at 4°C. The membrane was washed three times for 20 min each with TBS-T, followed by an 1 h incubation with the HRP-coupled streptavidin (1:10,000 in TBS-T) at RT. After three further washing steps with TBS-T for 20 min each, signals were detected by light emission with Pierce™ enhanced chemiluminescence reagent (ECL) western blot analysis substrate.

## Immunofluorescence analysis of fibroblasts

Control and patient fibroblasts ($2 \times 10^4$) were grown on glass coverslips in 24-well plates for 24 h and treated as described (Hanßke *et al*, 2002). Cells were stained with antibodies against GFUS (Thermo Fisher, PA5-27473, rabbit anti-human) and PMM2 (Abnova, H00005373-A01, mouse anti-human) in a dilution of 1:350, respectively. As secondary fluorochrome-conjugated antibodies, we used Alexa Fluor 488 (Thermo Fisher Scientific; goat anti-rabbit, 1:500) and Alexa Fluor 568 (Thermo Fisher Scientific; goat anti-mouse, 1:500). After washing with PBS, the cells were mounted in Fluoromount (DAKO Diagnostika GmbH, Hamburg, Germany) and analysed using a confocal laser scanning microscope (LSM3; Leica, Wetzlar, Germany).

## N-glycan analysis

N-glycan analysis was conducted by glyXboxCE™-system (glyXera, Magdeburg, Germany), based on multiplexed capillary gel electrophoresis with laser-induced fluorescence detection (xCGE-LIF), according to Hennig *et al* (2015) and Hennig *et al* (2016). Briefly, 2 µl of serum derived from controls ($n = 10$; age-matched) and the

### The paper explained

#### Problem
For many years, the number of known glycosylation defects is increasing, but there is still a large number of unknown or unexplained diseases in this field of protein modification. Therapy and healing approaches are extremely difficult to find, and for most CDG patients, no treatment is available.

#### Results
A wide range of patient diagnostics and comprehensive biochemical analyses led to the identification of the index patient with GFUS-CDG, a defect localized in the salvage pathway for GDP-L-fucose. Clinical symptoms of this patient were compared with clinical characteristics of other fucosylation deficiencies. The oral sugar treatment with L-fucose widely alleviated the symptomatic consequences of the GFUS-CDG patient.

#### Impact
Our study adds a new defect within the protein fucosylation pathway and will contribute to the identification and treatment of additional patients with L-fucose in the future.

patient were used for the analysis. N-glycan release with PNGase F, ′fluorescent labelling of N-glycans with 8-aminopyrene-1,3,6-trisulfonic acid (APTS) and the following postderivatization clean up by hydrophilic interaction liquid chromatography-solid phase extraction (HILIC-SPE) was carried out with the glyXprep16™ kit (glyXera, Magdeburg, Germany). Data processing, normalization of migration times and annotation of N-glycan fingerprints were performed with glyXtool™ software (glyXera, Magdeburg, Germany).

## Oral treatment with L-fucose

Food grade L-fucose (Jennewein Biotechnologie GmbH, Rheinbreitbach, Germany) was given starting with 87.5 mg/kg/day divided into three portions and weekly doubled to a final dose of 700 mg/kg/daily in three portions from day 21 onwards.

## FACS lectin analysis

Isolated PBMCs ($1 \times 10^6$ cells/staining) were centrifuged with $300 \times g$ for 5 min at 4°C. The cell pellet was resuspended in PBS, PBS containing 1 µg/ml fluorescein *Aleuria aurantia* (FAA) lectin or PBS containing FAA lectin + 200 mM L-fucose. PBMCs were incubated for 45 min on ice in the dark. After incubation, the cells were pelleted with $300 \times g$ for 5 min at 4°C and washed with PBS ($300 g$ for 5 min at 4°C). The cells were resuspended in 500 µl PBS. Cell staining was excited with a blue laser (488 nm), and the emission of fluorescein was detected via the FL1 channel by a flow cytometer (Cytomics FC 500 flow cytometer; Beckman Coulter, Brea, CA, USA). The data were analysed using Kaluza 1.3 (Beckman Coulter).

# Data availability

The here published material and data will be made available in accordance with the relevant ethical standards and legal guidelines.

Publication of the complete exome data is not included in the consent for clinical exome sequencing. Requests to access the datasets in more detail can be directed to the corresponding author.

**Expanded View** for this article is available online.

## Acknowledgements
Supported by the Austrian Science Fund (FWF) I4704-B and the Anniversary Fund of the Oesterreichische Nationalbank grant 18023 to S.B.W. and the Deutsche Forschungsgemeinschaft (DFG, German Research Foundation; Forschungsgruppe FOR 2509, Project-ID TH1461/7–2 to C.T. and RA2992/1-2 to E.R.) and the European Commission (E-Rare-3 Joint Transnational Call 2018/ EUROGLYCANOMICS in association with the DFG, Project-ID TH1461/9–1). We thank Ann-Katrin Schlosser for her outstanding contribution to this project and Dr. Marcus Hoffmann (MPI Magdeburg) for his excellent guidance and support during the assessment of the N-glycan analysis data. Graphical abstract was created with BioRender.com.

## Author contributions
JAM, CT, and SBW conceptualized and designed the study. RGF, AH, VG, AK, JAM, ER, CT, KB, DK, PH, and SBW acquired and analysed data. RGF, AH, CT, JAM, and SBW drafted the manuscript. All authors critically reviewed the manuscript and agree to the final version.

## Conflict of interest
ER is co-affiliated to Max Planck Institute for Dynamics of Complex Technical Systems and glyXera. All other authors declare that they have no conflict of interest.

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
