## [Review Process File · EMBO Molecular Medicine]

A spoonful of L-fucose - An efficient therapy for GFUS-CDG, a new glycosylation disorder

René Feichtinger, Andreas Hüllen, Andreas Koller, Dieter Kotzot, Valerian Grote, Erdmann Rapp, Peter Hofbauer, Karin Brugger, Christian Thiel, Johannes Mayr, and Saskia Wortmann

DOI: [10.15252/emmm.202114332](https://doi.org/10.15252/emmm.202114332)

Corresponding author: Saskia Wortmann (s.wortmann@salk.at)

Review Timeline:

Submission Date:	29th Mar 21
Editorial Decision:	11th May 21
Revision Received:	1st Jul 21
Editorial Decision:	21st Jul 21
Revision Received:	26th Jul 21
Accepted:	28th Jul 21

Editor: Jingyi Hou

Transaction Report:

11th May 2021

Dear Dr. Wortmann,

Thank you again for submitting your work to EMBO Molecular Medicine. We have now heard back from the four referees who evaluated your manuscript. As you will see from the reports below, the referees acknowledge the potential interest of the study. However, they also raise a series of concerns about your work, which should be convincingly addressed in a major revision of the present manuscript.

The referees' recommendations are rather clear and there is no need to reiterate their comments. All issues raised by the referees would need to be convincingly addressed. In particular, during our pre-decision cross-commenting process (in which the referees are given a chance to make additional comments, including on each other's reports), Referees #1 and #2 agreed with Referee #4's suggestion to increase the number of control samples.

We would welcome the submission of a revised version within three months for further consideration. Please note that EMBO Molecular Medicine strongly supports a single round of revision. As acceptance or rejection of the manuscript will depend on another round of review, your responses should be as complete as possible.

We are aware that many laboratories cannot function at full efficiency during the current COVID-19/SARS-CoV-2 pandemic and have therefore extended our "scooping protection policy" to cover the period required for a full revision to address the experimental issues. Please let me know should you need additional time, and also if you see a paper with related content published elsewhere.

I look forward to receiving your revised manuscript.

Jingyi

Jingyi Hou
Editor
EMBO Molecular Medicine

***** Reviewer's comments *****

Referee #1 (Comments on Novelty/Model System for Author):

The study described a novel CDG caused by GDP-L-fucose deficiency (GFUS mutation) in a child with developmental disability, growth retardation and dysmorphic facial features. GFUS protein and GDP-L-fucose were reduced leading to glycoprotein hypofucosylation. Moreover, the study shows that the conversion of GDP-D-[2- 3H]mannose to GDP-L-[2- 3H]fucose was dramatically reduced in patient-derived fibroblasts. Fucosylation was restored using retroviral complementation with the wildtype GFUS-cDNA of patient-derived fibroblasts which ascertained the causative effects of the variants detected.

The study well describes the effect of L-fucose supplementation on protein fucosylation in vitro and in vivo.

Referee #1 (Remarks for Author):

The study described a novel CDG caused by GDP-L-fucose deficiency (GFUS mutation) in a child with developmental disability, growth retardation and dysmorphic facial features. GFUS protein and GDP-L-fucose in patient-fibroblasts were reduced. Moreover, the study shows that the conversion of GDP-D-[2- 3H]mannose to GDP-L-[2- 3H]fucose was also dramatically reduced in patient-derived fibroblasts. Fucosylation was restored using retroviral complementation with the wildtype GFUS-cDNA of patient-derived fibroblasts which ascertained the causative effects of the variants detected.-

The study well describes the effect of L-fucose supplementation on protein fucosylation in vitro and in vivo.

Fucose supplementation restored fucosylation in patient-derived fibroblasts stained with AAL lectin.

The study interestingly shows that oral L-fucose supplementation was followed by improved fucosylation of proteins and some clinical improvements. Overall the study harbors a considerable novelty in the field and deserves particular interest in the light of the explored therapeutic approach.

Revisions required

1. The introduction briefly reports some essential elements of glycobiology with particular regard to the biosynthesis of GDP-L-fucose which improves comprehension to the non-specialist readers. However, information on CDG first-line screening test (i.e. serum transferrin IEF) and related pitfalls is particularly envisaged at this point.

Actually, this novel CDG adds to the list of CDG with normal serum transferrin IEF.

2. When citing therapeutic strategies in known CDG, it would be interesting to briefly report the different circumstances that make useful some monosaccharides as potential treatments in CDG.

Patient description

- I understand that reported prenatal information are related to fetal (child?) growth and fetal movements.

- Definition of "cerebral cysts" detected prenatally is vague and not sufficiently documented (i.e. localization, evolution during pregnancy are some information that should be added). -- Additional information on post-natal brain appearance as detected by newborn cerebral ultrasound might be meaningful also in the light of the increased head circumference at birth (97th pc).
- As the child was affected with a neurodevelopmental disorder, some information on possible seizure disorder and patient electroclinical features should be reported in the patient clinical description.

Biochemical and molecular investigations

- the mRNA expression levels for the Golgi GDPfucose transporter SLC35C1 (54.4% {plus minus} 6.5%; $p=***0.00003$) as well as for the potential ER GDP-fucose transporter SLC35C2 (63.3% {plus minus} 17.4%; $p=*0.035$ (Fig.4B)) were significantly diminished. Is there any explanation for this finding? This is particularly important to address the use of sugar supplementation as well.
- The effect of L-fucose supplementation on O-fucosylation is not adequately described in the results section. Please underline the strategies used to detect baseline deficiency and post-fucose modification of O-fucosylated proteins.

Clinical course after 19 months of oral L-fucose supplementation

- The clinical course was monitored by growth parameters and standardized psychological tests. It would be beneficial to know if CDG severity scoring was evaluated before and during treatment (i.e. Nijmegen CDG rating scale).
- Although there was a considerable improvement in mental capacities on psychological tests, it would be interesting to add some information on modification of age-related activities of daily living, if any.

Referee #2 (Comments on Novelty/Model System for Author):

Regarding the TECHNICAL QUALITY, my major concerns are about the xCGE-LIF analysis of N-glycome and FACS analysis of fucosylation of PBMCs.

- For these two points, although the presented data are finally rather convincing, corresponding (supplemental) Figure legends are overall unclear and very difficult to understand.

More precisely,

For xCGE-LIF,

- the number of controls (n=2) is unsatisfactory.
- thus, there is no corresponding statistical evaluation of results.
- nomenclature of N-glycans (A? G? D?...) should be clarified
- the claimed shift in the relative composition of the N-glycome is unclear viewing the 'difficult' Table EV1 (and not 'supplementary table 2' as mentioned?)
- why the announced increase of hybrid-type N-glycans not linked to detectable transferrin glycosylation defects?

For FACS analysis of the fucosylation of PBMCs,

- references to figure 5 and to suppl Fig3 (3 pages) seem wrong and are confusing. For example, fig 5C (8 weeks fucose) is mentioned for 2 weeks fucose in the Fig legend?

- Legend of Fig EV3 is very unclear

In addition,

FigEV4 is never mentioned in the manuscript? Is there a missing page in the submitted text?

Regarding the Fig. 5A (AAL lectin blot), I'm surprised that only 3 protein bands (what MW?) seem fucosylated in serum... Furthermore, legends of fig 5B/5C should be clarified.

NOVELTY is high since Feichtinger and col. described here a new treatable CDG.

MEDICAL IMPACT could be rather high (medium?) for this probably treatable CDG that concerns very few affected individuals. But evaluated clinical test(s) are rather scarce with rather slight improvements...

MODEL SYSTEM (fibroblasts, serum, platelets, PBMCs and the affected girl) are adequate.

Referee #2 (Remarks for Author):

Review of the manuscript entitled: A spoonful of L-fucose - An efficient therapy for GFUS-CDG, a new glycosylation disorder, by Feichtinger and col.

This paper describing a new potentially treatable CDG is interesting (high novelty and potential medical impact for affected individuals) and well-documented (technical issues; literature) but needs to be strongly clarified and corrected.

In addition, I think that a more concise version should be proposed as a short report in EMBO Molecular Medicine.

MAJOR REMARKS:

My major concerns are about the xCGE-LIF analysis of N-glycome and FACS analysis of fucosylation of PBMCs.

- For these two points, although most of the presented data (xCGE-LIF?) are finally rather convincing, corresponding (supplemental) figure legends are overall unclear and very difficult to follow and understand.

For xCGE-LIF,

- the number of controls (n=2) is unsatisfactory. Thus, there is no corresponding statistical evaluation of results.

- nomenclature of N-glycans (A? G? D?...) should be clarified

- the claimed shift in the relative composition of the N-glycome is not obvious viewing the 'difficult' Table EV1 (and not 'supplementary table 2' as mentioned?)

- why the announced increase of hybrid-type N-glycans not linked to detectable transferrin glycosylation defects?

For FACS analysis of the fucosylation of PBMCs,

- references to figure 5 and to suppl Fig3 (3 pages) seem wrong and are confusing. For example, fig 5C (8 weeks fucose) is mentioned for 2 weeks fucose in the Fig legend?

- Legend of Fig EV3 is very unclear

In addition,

FigEV4 is never mentioned in the manuscript? Is there a missing page in the submitted text?

Regarding Fig. 5A (AAL lectin blot), I'm surprised that only 3 protein bands (what MW?) seem fucosylated in serum... Furthermore, legends of fig 5B/5C should be clarified.

Concerning the text of the manuscript (essentially Discussion):

- I feel that the reader could be very confused between O-fucosylation and fucosylation of O-glycans (e.g. GalNAc O-glycans).
- Bombay group should be further explained. Unclear point.
- 'expression of SLC35C1 and FUT8 was found to be reduced in QRT-PCR studies' : where are the related data for FUT8? I did not find.
- Why the accumulation of FA3G3S1(2,6) (??) is interesting?
- 'expression of POFUT1 and POFUT2 were normal in qRT-PCR studies': where are the related data? I did not find.
- what about the described elevated protein level of FUT8?

MINOR POINTS:

- 'Kaufmann ABC' is unclear in the abstract.
- GAG (O-xyl) should be mentioned in the introduction.
- 'From birth on lack of appetite' (case reports): I don't understand.
- what were the tested prediction programs for gene variant pathogenicity?
- how long for 100 μ M L-fucose treatment? (results).
- ADAMTS13 should be clarified in the results part: ADAMTS13, a heavily O-fucosylated protein...
- 'This could be due to the functional redundancy of the salvage and the de novo pathway': unclear.

Referee #3 (Remarks for Author):

This is a well-written manuscript on a novel CDG.

I have no problem with the content. The following are a few small corrections:

ABSTRACT

L 4: delay, mild

L 5: synthase,

L 8: serum glycoproteins, leukocytes

L 12: serum glycoproteins

INTRODUCTION

L 1: Some 2 %

L 6: anchored protein synthesis

L 11, 12: guanosine (not guanidine!)

L 23: CDG (is singular as well as plural)

10th last sentence: CDG no therapeutic

RESULTS

P 2, L 17: GDP-D-mannose

eight last line: no side effects

Referee #4 (Remarks for Author):

The manuscript details the identification of compound heterozygous missense mutations in GFUS encoding GDP-L-fucose synthase. This led to a novel CDG. The authors also show that fucose supplementation was at least partially effective as a treatment. The manuscript is well written and concise. The two aspects of the work, namely i) the identification of a new CDG, GFUS-CDG, and ii) the clinical study on the effectiveness of treatment with fucose are valuable to the scientific and clinical community.

With regards to i) the authors present a convincing case reinforced by reduced GFUS protein levels and a greatly decreased activity of GFUS in patient fibroblasts, as measured by the conversion of GDP-D-[2-3H]mannose to GDP-L-[2-3H]fucose. In combination clinical data that shows a broadly similar picture to existing disorders of fucose metabolism, this leads the reviewer to conclude that the GFUS variants in the patient presented are indeed causative for their disorder.

With regards to ii) the data is also relatively convincing due to lectin staining and fucosylation results from patient samples, as well as limited improvement in the clinical course. So far, only one patient is described. The authors address briefly why there haven't been any other patients identified, because the GFUS^{-/-} mouse is nonviable and in this patient there is 1% residual activity, allowing some small amount of de novo fucose synthesis. More severe mutations would thus be lethal.

Major comments:

- For several data panels and results it seems only one control was used to compare with that of the patient (e.g. EV Fig 2.) Ideally two or several controls would be used for comparison, particularly for an inherently variable method such as western blotting, and bearing in mind only one patient was available so statistical analysis was already very difficult. If the authors in fact used several controls and the data shown was purely for presentation reasons, it should be stated clearly in the text. Particularly with regards to EV Fig 2, this makes it very hard to draw conclusions from the comparison of a single control with a single patient. This should at the very least be noted.

Minor comments:

- Abstract/introduction: The type of variant identified in the patients is not specified beyond 'biallelic' before the results section. The authors should add that the variants are compound heterozygous missense mutations to the abstract and/or introduction.
- In the discussion it is mentioned that when measured by qPCR, lower levels of transcripts from FUT8 were found. Firstly, this is not mentioned in results and should be added. Secondly, does this not oppose the higher FUT8 protein levels found by western blot? Can you explain this discrepancy?
- The authors also present qPCR data showing that GFUS, SLC35C1, SLC35C2 were all downregulated on a transcriptional level and describe this as a possible compensatory mechanism. Is there evidence of this in other defects of the fucose metabolic pathway?
- Do the authors think that in this case, with some residual enzymatic activity of GFUS, there is potential for supplementation with mannose (or analogue) as a substrate therapy? Or would this be unlikely to improve on the results already achieved by fucose therapy?
- Park et al. (<https://pubmed.ncbi.nlm.nih.gov/33312876/>) recently reported that fucose treatment was partially successful in a case of FUT8-CDG. Although their study was only on two patients and showed very tentative results, it is very relevant to this manuscript and should be at least

mentioned.

***** Reviewer's comments *****

REFEREE #1 (COMMENTS ON NOVELTY/MODEL SYSTEM FOR AUTHOR):

The study described a novel CDG caused by GDP-L-fucose deficiency (GFUS mutation) in a child with developmental disability, growth retardation and dysmorphic facial features. GFUS protein and GDP-L-fucose were reduced leading to glycoprotein hypofucosylation. Moreover, the study shows that the conversion of GDP-D-[2- 3H]mannose to GDP-L-[2- 3H]fucose was dramatically reduced in patient-derived fibroblasts. Fucosylation was restored using retroviral complementation with the wildtype GFUS-cDNA of patient-derived fibroblasts which ascertained the causative effects of the variants detected.

The study well describes the effect of L-fucose supplementation on protein fucosylation in vitro and in vivo.

First we want to thank all of the reviewers for their helpful comments and suggestions to improve the manuscript.

Referee #1 (Remarks for Author):

The study described a novel CDG caused by GDP-L-fucose deficiency (GFUS mutation) in a child with developmental disability, growth retardation and dysmorphic facial features. GFUS protein and GDP-L-fucose in patient-fibroblasts were reduced. Moreover, the study shows that the conversion of GDP-D-[2- 3H]mannose to GDP-L-[2- 3H]fucose was also dramatically reduced in patient-derived fibroblasts. Fucosylation was restored using retroviral complementation with the wildtype GFUS-cDNA of patient-derived fibroblasts which ascertained the causative effects of the variants detected.- The study well describes the effect of L-fucose supplementation on protein fucosylation in vitro and in vivo.

Fucose supplementation restored fucosylation in patient-derived fibroblasts stained with AAL lectin. The study interestingly shows that oral L-fucose supplementation was followed by improved fucosylation of proteins and some clinical improvements. Overall the study harbors a considerable novelty in the field and deserves particular interest in the light of the explored therapeutic approach.

Revisions required

1. The introduction briefly reports some essential elements of glycobiology with particular regard to the biosynthesis of GDP-L-fucose which improves comprehension to the non-specialist readers. However, information on CDG first-line screening test (i.e. serum transferrin IEF) and related pitfalls is particularly envisaged at this point. Actually, this novel CDG adds to the list of CDG with normal serum transferrin IEF.

- *We agree to this important point. Due to didactical reasons, we preferred to point this diagnostic issue out in the discussion part. We added the sentence 'Thus, GFUS-CDG adds to the list of glycosylation defects with normal serum transferrin IEF.' To the penultimate paragraph of the discussion on page 14.*

2. When citing therapeutic strategies in known CDG, it would be interesting to briefly report the different circumstances that make useful some monosaccharides as potential treatments in CDG.

- *We added the sentence 'Those therapies usually rely on a surplus of the respective donor substrate or co-factor for the defective protein.' In the introduction in the second last paragraph on page 4*

3. Patient description. I understand that reported prenatal information are related to fetal (child?) growth and fetal movements.

- *Definition of "cerebral cysts" detected prenatally is vague and not sufficiently documented (i.e. localization, evolution during pregnancy are some information that should be added). -- Additional information on post-natal brain appearance as detected by newborn cerebral ultrasound might be meaningful also in the light of the increased head circumference at birth (97th pc).*

4. As the child was affected with a neurodevelopmental disorder, some information on possible seizure disorder and patient electroclinical features should be reported in the patient clinical description.

- *According to the reviewer suggestion we included some information about the mentioned issues in the clinical description.*

5. Biochemical and molecular investigations. The mRNA expression levels for the Golgi GDP fucose transporter SLC35C1 (54.4% {plus minus} 6.5%; p=***0.00003) as well as for the potential ER GDP-fucose transporter SLC35C2 (63.3% {plus minus} 17.4%; p=*0.035 (Fig.4B)) were significantly diminished. Is there any explanation for this finding? This is particularly important to address the use of sugar supplementation as well.

- *We rate this as a compensatory effect due to a lack of the substrate, GDP-L-fucose, for both transporters. We stated this in the discussion on page 12 paragraph 7.*

6. The effect of L-fucose supplementation on O-fucosylation is not adequately described in the results section. Please underline the strategies used to detect baseline deficiency and post-fucose modification of O-fucosylated proteins.

- *O-fucosylation depends on a sufficient supply of GDP-L-fucose, therefore we measured the expression of POFUT1/2 on the transcript and protein level and included both datasets in the results and discussion parts. Furthermore, we used secretion of ADAMTS13 in the serum as an indirect marker for protein-O-fucosylation by POFUT2. In short, we found a reduced POFUT2 and ADAMTS13 protein amount. As secretion of ADAMTS13 depends on the O-fucosylation of its thrombospondin repeats catalyzed by POFUT2, we conclude that the diminished GDP-L-fucose level first led to reduced POFUT2 and subsequently impacts fucosylation and thereby secretion of ADAMTS13. The information can be found in the results part 'GFUS deficiency affects expression of downstream proteins associated with fucose metabolism' on page 8 and in the discussion*

on pages 12/13 paragraph 9. Also expanded view figure 2 was updated with those data including statistics and an increased amount of control samples.

7. Clinical course after 19 months of oral L-fucose supplementation. The clinical course was monitored by growth parameters and standardized psychological tests. It would be beneficial to know if CDG severity scoring was evaluated before and during treatment (i.e. Nijmegen CDG rating scale).

- *see reply to 8. below.*

8. Although there was a considerable improvement in mental capacities on psychological tests, it would be interesting to add some information on modification of age-related activities of daily living, if any.

- *We have added the Nijmegen Pediatric CDG Rating Scales to our manuscript (abstract, case description, supplementary data) and have provided more details on the activities of daily living to the follow up section.*

Referee #2 (Comments on Novelty/Model System for Author):

1. Regarding the TECHNICAL QUALITY, my major concerns are about the xCGE-LIF analysis of N-glycome and FACS analysis of fucosylation of PBMCs. For these two points, although the presented data are finally rather convincing, corresponding (supplemental) Figure legends are overall unclear and very difficult to understand.

- *According to the reviewer suggestion we included more detailed figure legends for the xCGE LIF analysis and FACS analysis.*

More precisely,

For xCGE-LIF,

- the number of controls (n=2) is unsatisfactory.

- thus, there is no corresponding statistical evaluation of results.

- *We increased the number of controls to n=10. SD for the controls was added to the result part 'Normal behaviour of serum transferrin during isoelectric focusing but abnormal whole serum N-glycosylation'. Expanded view figure 1 was adapted as well to increase clarity.*

- nomenclature of N-glycans (A? G? D?...) should be clarified

- *The respective information was added to the figure legend of expanded view figure 1 (N-glycan structures were produced using Glycan Builder 2 (Tsuchiya, Aoki et al., 2017) following the Symbol Nomenclature for Glycans (SNFG) guidelines (Neelamegham, Aoki-Kinoshita et al., 2019). Naming of N-glycan structures was adapted from the Oxford nomenclature (Doherty, Theodoratou et al., 2018)).*

- the claimed shift in the relative composition of the N-glycome is unclear viewing the 'difficult' Table EV1 (and not 'supplementary table 2' as mentioned?)

- *Regarding the N-glycome composition, Data presentation of the results was reevaluated and can be found in a clarified version of expanded view figure 1 and 'Expanded View Table 1: N-Glycans identified and quantified by xCGE-LIF'.*

- why the announced increase of hybrid-type N-glycans not linked to detectable transferrin glycosylation defects?

- *Transferrin displays the gold marker in CDG diagnostics. Unfortunately, it is not a valuable tool for detection of hybrid type glycans as the overwhelming part of N-glycans bound to transferrin belong to the complex type. Hybrid glycans make up only about 4-5% and thus have a rather small influence on the transferrin glycosylation status.*

For FACS analysis of the fucosylation of PBMCs,

- references to figure 5 and to suppl Fig3 (3 pages) seem wrong and are confusing. For example, fig 5C (8 weeks fucose) is mentioned for 2 weeks fucose in the Fig legend?

- *We corrected and modified figure legend 5. We are sorry we were not able to figure out what is wrong with 8 weeks and 2 weeks.*

- Legend of Fig EV3 is very unclear

- *We modified the legends for the FACS part for figure legend EV3.*

In addition,

FigEV4 is never mentioned in the manuscript? Is there a missing page in the submitted text?

We included Fig EV4 in the text.

- *A reference to FigEV4 was included.*

Regarding the Fig. 5A (AAL lectin blot), I'm surprised that only 3 protein bands (what MW?) seem fucosylated in serum... Furthermore, legends of fig 5B/5C should be clarified.

- *Concerning the AAL blot of control and patient serum (Fig. 5A), we increased brightness and contrast to clarify that there are more protein bands than three.*
- *We modified figure legend 5.*

NOVELTY is high since Feichtinger and col. described here a new treatable CDG.

MEDICAL IMPACT could be rather high (medium?) for this probably treatable CDG that concerns very few affected individuals. But evaluated clinical test(s) are rather scarce with rather slight improvements...

MODEL SYSTEM (fibroblasts, serum, platelets, PBMCs and the affected girl) are adequate.

Concerning the text of the manuscript (essentially Discussion):

- I feel that the reader could be very confused between O-fucosylation and fucosylation of O-glycans (e.g. GalNAc O-glycans).

- *We removed this inconsistency.*

- Bombay group should be further explained. Unclear point.

- *We added this sentence to the manuscript: People who carry no functional H antigens on their red blood cells (e.g. due to hypofucosylation), present serologically with the Bombay (Oh) blood group.*

- 'expression of SLC35C1 and FUT8 was found to be reduced in QRT-PCR studies' : where are the related data for FUT8? I did not find.

- *Expression data for SLC35C1, SLC35C2, FUT8, POFUT1 and POFUT2 were included in the results and discussion part and in figure 4 and expanded view figure 2. Please also see our answer to reviewer 1 above*

- Why the accumulation of FA3G3S1(2,6) (??) is interesting?

- *By using an increased number of controls in combination with more exoglycosidases, we were able to decrease the number of undistinguishable N-glycan structures, which led to the identification of a specific core-fucosylated glycan (FA2G2S1(2,6)S1(2,3), which might be interesting as it could be used as a biomarker for GFUS-CDG diagnosis.*

- 'expression of POFUT1 and POFUT2 were normal in qRT-PCR studies': where are the related data? I did not find. – *please see above*

- what about the described elevated protein level of FUT8?

- *Please see above*

MINOR POINTS:

- 'Kaufmann ABC' is unclear in the abstract.

- *The Kaufmann ABC is a standardised and world-wide widely used test battery. It helps to identify an individual's strengths and weaknesses in cognitive ability and mental processing. The information provided by the KABC-II can facilitate clinical and educational planning, treatment planning and placement decisions. The internal consistency reliability coefficient for core and supplementary subtests demonstrate the KABC-II has good reliability. We would therefore think that this does not need more clarification.*

- GAG (O-xyI) should be mentioned in the introduction.

- *o-xyl was added to the first paragraph of the introduction*
- 'From birth on lack of appetite' (case reports): I don't understand.
 - *She did not indicate when she was hungry, neither with crying for breastfeeding in the first months, neither by showing any interest in food later on. This improved after the start of L-fucose and is now detailed more precisely in the follow up section.*
- what were the tested prediction programs for gene variant pathogenicity?
 - *We added SIFT, PolyPhen, REVEL, MetaLR to the manuscript (see Results 'Biallelic variants in GFUS', page 5/6).*
- how long for 100 μM L-fucose treatment? (results).
 - *L-fucose treatment was done for 48 h and this information was added to the method- and results part.*
- ADAMTS13 should be clarified in the results part: ADAMTS13, a heavily O-fucosylated protein.
 - *We added this information to the respective paragraph 'GFUS deficiency affects expression of downstream proteins associated with fucose metabolism' in the results.*
- 'This could be due to the functional redundancy of the salvage and the de novo pathway': unclear.
 - *We replaced 'redundancy' by 'overlap' to be more precise on the meaning of the sentence.*

Referee #3 (Remarks for Author):

This is a well-written manuscript on a novel CDG.

I have no problem with the content. The following are a few small corrections:

ABSTRACT

L 4: delay, mild

- *Corrected.*

L 5: synthase,

- *Corrected.*

L 8: serum glycoproteins, leukocytes

- *Corrected.*

L 12: serum glycoproteins

- *Corrected.*

INTRODUCTION

L 1: Some 2 %

- *Corrected.*

L 6: anchored protein synthesis

- *Corrected.*

L 11, 12: guanosine (not guanidine!)

- *Corrected.*

L 23: CDG (is singular as well as plural)

- *Corrected.*

10th last sentence: CDG no therapeutic

- *Corrected.*

RESULTS

P 2, L 17: GDP-D-mannose

- *Corrected.*

eight last line: no side effects

- *Corrected.*

Referee #4 (Remarks for Author):

The manuscript details the identification of compound heterozygous missense mutations in GFUS encoding GDP-L-fucose synthase. This led to a novel CDG. The authors also show that fucose supplementation was at least partially effective as a treatment. The manuscript is well written and concise. The two aspects of the work, namely i) the identification of a new CDG, GFUS-CDG, and ii) the clinical study on the effectiveness of treatment with fucose are valuable to the scientific and clinical community.

With regards to i) the authors present a convincing case reinforced by reduced GFUS protein levels and a greatly decreased activity of GFUS in patient fibroblasts, as measured by the conversion of GDP-D-[2-3H]mannose to GDP-L-[2-3H]fucose. In combination clinical data that shows a broadly similar picture to existing disorders of fucose metabolism, this leads the reviewer to conclude that the GFUS variants in the patient presented are indeed causative for their disorder.

With regards to ii) the data is also relatively convincing due to lectin staining and fucosylation results from patient samples, as well as limited improvement in the clinical course. So far, only one patient is described. The authors address briefly why there haven't been any other patients identified, because the GFUS^{-/-} mouse is nonviable and in this patient there is 1% residual activity, allowing some small amount of de novo fucose synthesis. More severe mutations would thus be lethal.

Major comments:

- For several data panels and results it seems only one control was used to compare with that of the patient (e.g. EV Fig 2.) Ideally two or several controls would be used for comparison, particularly for an inherently variable method such as western blotting, and bearing in mind only one patient was available so statistical analysis was already very difficult. If the authors in fact used several controls and the data shown was purely for presentation reasons, it should be stated clearly in the text. Particularly with regards to EV Fig 2, this makes it very hard to draw conclusions from the comparison of a single control with a single patient. This should at the very least be noted.

- *We added more controls to the experiments shown in EV Fig 2 and EV Fig 1 and also included the data to the results and discussion part. Please see also our comments to reviewer #1.*

Minor comments:

- Abstract/introduction: The type of variant identified in the patients is not specified beyond 'biallelic' before the results section. The authors should add that the variants are compound heterozygous missense mutations to the abstract and/or introduction.

- *We included the required information in the abstract line 4: "... we detected biallelic variants in GFUS (NM_003313.4) c.[632G>A];[659C>T] (p.[Gly211Glu];[Ser220Leu]) in a patient presenting with global developmental delay ..."*

- In the discussion it is mentioned that when measured by qPCR, lower levels of transcripts from FUT8 were found. Firstly, this is not mentioned in results and should be added. Secondly, does this not oppose the higher FUT8 protein levels found by western blot? Can you explain this discrepancy?

- *We thank the reviewer to point us to a mistake made in our manuscript: The transcript and protein level of FUT8 were both found to be increased in case of the patient. We apologize for this confusion. The error was corrected and classified accordingly in the discussion (see Discussion paragraph 8, page 11).*

- The authors also present qPCR data showing that GFUS, SLC35C1, SLC35C2 were all downregulated on a transcriptional level and describe this as a possible compensatory mechanism. Is there evidence of this in other defects of the fucose metabolic pathway?

- *To the best of our knowledge no qPCR data on GFUS and the transporters have been published so far.*

- Do the authors think that in this case, with some residual enzymatic activity of GFUS, there is potential for supplementation with mannose (or analogue) as a substrate therapy? Or would this be unlikely to improve on the results already achieved by fucose therapy?

- *Thanks for this question, which is not only of interest of the biochemical but as well of the financial point of view, as the costs for mannose are about 1000€ per kg and for fucose are about 7000€ per kg. Regrettably, measurements of nucleotide activated sugars in patient derived fibroblasts (data not shown) already showed an increase in GDP-mannose compared to control cells, which led us to the assumption that further supplementation of mannose would have no additional effect as a substrate therapy.*

- Park et al. (<https://pubmed.ncbi.nlm.nih.gov/33312876/>) recently reported that fucose treatment was partially successful in a case of FUT8-CDG. Although their study was only on two patients and showed very tentative results, it is very relevant to this manuscript and should be at least mentioned.

- *The publication of Park et al. on FUT8-CDG is included in the discussion on page 10.*

21st Jul 2021

Dear Dr. Wortmann,

Thank you for the submission of your revised manuscript to EMBO Molecular Medicine. We have now received the enclosed reports from the two referees who agreed to re-assess it. As you will see the referees are now supportive and I am pleased to inform you that we will be able to accept your manuscript pending the following amendments:

***** Reviewer's comments *****

Referee #1 (Comments on Novelty/Model System for Author):

The current version of the manuscript has carefully taken into account all my previous revisions. The revised manuscript is clear and complete and it deserves to be published in EMBO Molecular Medicine in the opinion of this reviewer.

Referee #1 (Remarks for Author):

The current version of the manuscript has carefully taken into account all my previous revisions. The revised manuscript is clear and complete. I have not further comments for the present study.

Referee #2 (Remarks for Author):

All my comments and corrections have been fulfilled.
Nice, clear, and very interesting work!
Thank you!

The authors performed the requested editorial changes.

28th Jul 2021

Dear Dr. Wortmann,

We are pleased to inform you that your manuscript is accepted for publication and is now being sent to our publisher to be included in the next available issue of EMBO Molecular Medicine.

We would like to remind you that as part of the EMBO Publications transparent editorial process initiative, EMBO Molecular Medicine will publish a Review Process File online to accompany accepted manuscripts. If you do NOT want the file to be published or would like to exclude figures, please immediately inform the editorial office via e-mail.

Please read below for additional IMPORTANT information regarding your article, its publication and the production process.

Congratulations on your interesting work,

Kind regards,
Jingyi

Jingyi Hou
Editor
EMBO Molecular Medicine

Follow us on Twitter @EmboMolMed
Sign up for eTOCs at embopress.org/alertsfeeds

*** ** IMPORTANT INFORMATION ** **

SPEED OF PUBLICATION

The journal aims for rapid publication of papers, using the advance online publication "Early View" to expedite the process: A properly copy-edited and formatted version will be published as "Early View" after the proofs have been corrected. Please help the Editors and publisher avoid delays by providing e-mail address(es), telephone and fax numbers at which author(s) can be contacted.

Should you be planning a Press Release on your article, please get in contact with embomolmed@wiley.com as early as possible, in order to coordinate publication and release dates.

LICENSE AND PAYMENT:

All articles published in EMBO Molecular Medicine are fully open access: immediately and freely available to read, download and share.

EMBO Molecular Medicine charges an article processing charge (APC) to cover the publication

costs. You, as the corresponding author for this manuscript, should have already received a quote with the article processing fee separately. Please let us know in case this quote has not been received.

Once your article is at Wiley for editorial production you will receive an email from Wiley's Author Services system, which will ask you to log in and will present you with the publication license form for completion. Within the same system the publication fee can be paid by credit card, an invoice, pro forma invoice or purchase order can be requested.

Payment of the publication charge and the signed Open Access Agreement form must be received before the article can be published online.

PROOFS

You will receive the proofs by e-mail approximately 2 weeks after all relevant files have been sent to our Production Office. Please return them within 48 hours and if there should be any problems, please contact the production office at embopressproduction@wiley.com.

Please inform us if there is likely to be any difficulty in reaching you at the above address at that time. Failure to meet our deadlines may result in a delay of publication.

All further communications concerning your paper proofs should quote reference number EMM-2021-14332-V3 and be directed to the production office at embopressproduction@wiley.com.

Thank you,

Jingyi Hou
Editor
EMBO Molecular Medicine

Corresponding Author Name: Saskia B. Wortmann

Journal Submitted to: EMBO Mol Med

Manuscript Number: EMM-2021-14332-V2